# Grafting nanometer metal/oxide interface towards enhanced low-temperature acetylene semi-hydrogenation

Shihui Zou [1,11✉], Baohui Lou[1,11], Kunran Yang [2,11], Wentao Yuan[3], Chongzhi Zhu[4], Yihan Zhu [4✉], Yonghua Du [5,6✉], Linfang Lu[1], Juanjuan Liu [7], Weixin Huang [8], Bo Yang [2✉], Zhongmiao Gong[9], Yi Cui [9], Yong Wang[3], Lu Ma[6], Jingyuan Ma[10], Zheng Jiang [10], Liping Xiao[1] & Jie Fan [1✉]

Metal/oxide interface is of fundamental significance to heterogeneous catalysis because the seemingly "inert" oxide support can modulate the morphology, atomic and electronic structures of the metal catalyst through the interface. The interfacial effects are well studied over a bulk oxide support but remain elusive for nanometer-sized systems like clusters, arising from the challenges associated with chemical synthesis and structural elucidation of such hybrid clusters. We hereby demonstrate the essential catalytic roles of a nanometer metal/oxide interface constructed by a hybrid $Pd/Bi_2O_3$ cluster ensemble, which is fabricated by a facile stepwise photochemical method. The $Pd/Bi_2O_3$ cluster, of which the hybrid structure is elucidated by combined electron microscopy and microanalysis, features a small Pd-Pd coordination number and more importantly a Pd-Bi spatial correlation ascribed to the heterografting between Pd and Bi terminated $Bi_2O_3$ clusters. The intra-cluster electron transfer towards Pd across the as-formed nanometer metal/oxide interface significantly weakens the ethylene adsorption without compromising the hydrogen activation. As a result, a 91% selectivity of ethylene and 90% conversion of acetylene can be achieved in a front-end hydrogenation process with a temperature as low as 44 °C.

[1] Key Lab of Applied Chemistry of Zhejiang Province, Department of Chemistry, Zhejiang University, 310027 Hangzhou, China. [2] School of Physical Science and Technology, ShanghaiTech University, 201210 Shanghai, China. [3] School of Materials Science and Engineering, Zhejiang University, 310027 Hangzhou, China. [4] Center for Electron Microscopy, State Key Laboratory Breeding Base of Green Chemistry Synthesis Technology and College of Chemical Engineering, Zhejiang University of Technology, 310014 Hangzhou, China. [5] Institute of Chemical and Engineering Sciences, A*STAR, 627833 Singapore, Singapore. [6] National Synchrotron Light Source II, Brookhaven National Laboratory, Upton, NY 11973, USA. [7] College of Materials & Environmental Engineering, Hangzhou Dianzi University, 310036 Hangzhou, China. [8] Department of Chemical Physics, University of Science and Technology of China, 230026 Hefei, China. [9] Vacuum Interconnected Nanotech Workstation, Suzhou Institute of Nano-Tech and Nano-Bionics, Chinese Academy of Sciences, 215123 Suzhou, China. [10] Shanghai Synchrotron Radiation Facility, Shanghai Institute of Applied Physics Chinese Academy of Sciences, 201800 Shanghai, China. [11] These authors contributed equally: Shihui Zou, Baohui Lou, Kunran Yang. ✉email: xueshan199@163.com; yihanzhu@zjut.edu.cn; ydu@bnl.gov; yangbo1@shanghaitech.edu.cn; jfan@zju.edu.cn

Metal/oxide interface is of great fundamental and practical interest to heterogeneous catalysis, because it raises essential questions regarding the strong metal–support interaction[1,2] and plays a pivotal role in several catalytic processes[3,4]. From a structural perspective, a metal/oxide interface is constructed by components that significantly differ from each other in terms of chemical compositions, bonding characters, lattice parameters, and electric and mechanical properties[5,6], of which the adhesion structure and chemistry turn out to be a compelling research topic, while from a functional perspective, the chemical bonding and associated charge transfer[7] at the metal/oxide interface allow the modulations of morphology, size, and electronic structures of metals to optimize the bonding strength of reaction intermediates for better catalytic performances[3]. During the past few decades, considerable progress has been achieved in the structural elucidation and tuning of well-defined metal/oxide interfaces that usually adopt a bulk oxide support to facilitate the nucleation, adsorption, or deposition of metals[8]. It is expected that a nanometer metal/oxide interface, perhaps formed by heterografting between metal and oxide clusters, would reinforce the structural and electronic effect to achieve better catalytic performance. However, due to the great challenges in the chemical synthesis and structural elucidation of such hybrid clusters, there are limited insights into the nanometer metal/oxide interface.

As a representative reaction where oxides supported metal catalysts are frequently used, the selective hydrogenation of acetylene to ethylene is subjected to an inherent trade-off between two essential requirements for both high catalytic activity and selectivity: the facile activation of hydrogen and the weak binding of ethylene[9]. Despite the significant progress achieved by Pd-based catalysts[10], a simultaneous optimization of these two parameters is still challenging, especially in the front-end process where $H_2$ and $C_2H_4$ are in large excess. To reach this goal requires a sophisticated tuning of the geometric and electronic structure of Pd, which motivates people to engineer the metal/oxide interface. In most Pd/oxide catalysts, only Pd nanoparticles or isolated Pd atoms are loaded. Unfortunately, Pd nanoparticles are efficient to activate hydrogen at low temperatures, but their strong binding with ethylene favors sequential hydrogenation of ethylene to ethane[10,11]. Isolated Pd site catalysts including Pd single-atom catalysts[12–15] and Pd-based intermetallic compounds[16–22] feature weak π-bonding with ethylene and thus good selectivity in the acetylene hydrogenation reaction, but their concomitant weakened hydrogen activation requires a relatively high reaction temperature (>100 °C) to achieve high conversion of acetylene, which potentially leads to a safety concern in the reactor beds[21]. Decreasing the size of oxide supports to nanocluster scale would remarkably change their coordination number (CN)[23], surface termination[24], and d-band character[25], which allow a strong chemical and electronic interaction with Pd to continuously regulate the size and electronic structure of Pd. Among them, ligand-free Pd clusters stabilized by a nanometer metal/oxide interface are expected to bridge the size and performance gaps between Pd nanoparticles and single atoms with maximized interfacial effects.

Herein we propose a facile stepwise photochemical strategy to fabricate hybrid $Pd/Bi_2O_3$ clusters where the Pd clusters are stabilized by the ~1 nm ordered $Bi_2O_3$ cluster through the nanometer interfaces. These hybrid clusters are well and stably dispersed on $TiO_2$ substrate with a high Pd loading up to 2.3 wt.%. They exhibit a low Pd–Pd CN of 4.7 and more importantly a Pd–Bi spatial correlation ascribed to the heterografting between Pd and Bi-terminated $Bi_2O_3$ clusters. Interestingly, these hybrid clusters feature an intra-cluster electron transfer toward Pd and result in a deeper d-band center compared with other Pd metals,

which enables much weaker ethylene binding without compromising the hydrogen activation activity. As a result, a 90% conversion of acetylene together with a 91% selectivity to ethylene is achieved in excess of ethylene and at a temperature as low as 44 °C.

## Results and discussion

**Synthesis and structural characterizations.** $Pd_{1.0}/Bi_2O_3/TiO_2$ catalysts were prepared by a stepwise photo-deposition of Bi and Pd on $TiO_2$ with a molar ratio of Bi/Pd kept at 1.0 (please see details in "Methods"). In brief, $Bi^{3+}$ was reduced by the photo-generated electrons to produce $Bi^0$ clusters on $TiO_2$. Subsequently, Pd was deposited onto $Bi/TiO_2$ to form $Pd/Bi/TiO_2$. When $Bi/TiO_2$ and $Pd/Bi/TiO_2$ were exposed to the air, Bi was spontaneously oxidized into $Bi_2O_3$ because the Gibbs free energy of the reaction is minus[26]. The synthetic procedure is schematically illustrated in Fig. 1a. Inductively coupled plasma–atomic emission spectroscopy (ICP-AES) suggests that the mass loading of Bi and Pd are 4.9 wt.% and 2.3 wt.%, respectively, close to the feeding ratio. Our previous study has proved that $TiO_2$ has a strong interaction with $Bi^{3+}$, characterized by an unprecedented 1.5 eV upshift of Ti $2p$[27]. Such interaction ensures the formation of highly dispersed Bi on $TiO_2$ during the following reduction of $Bi^{3+}$[28]. As shown in Supplementary Fig. 1a, b, Bi species are uniformly dispersed on $TiO_2$ as ~1 nm clusters with a Bi loading up to 5 wt.%. Interestingly, close inspection of individual Bi cluster on the $TiO_2$ support by using aberration-corrected annular dark-field scanning transmission electron microscopy (ADF-STEM) indicates that the Bi cluster has an ordered α-$Bi_2O_3$ structure with a highly distorted lattice, which exhibits relatively weak diffuse peaks in the fast Fourier transform (FFT; Fig. 1b). This is consistent with the literature result[26], which indicates that the mild oxidation of $Bi^0$ in the air is thermodynamically spontaneous and usually forms monoclinic α-$Bi_2O_3$ (monoclinic, $P2_1/c(14)$). In addition to the intrinsic lattice distortion that blurs the image contrast, the beam-induced structural dynamics of the highly beam-sensitive $Bi_2O_3$ clusters may further introduce image blurring, mainly arising from the remarkable knock-on displacements and radiolysis. Despite these beam-induced effects, careful inspection on these ADF-STEM images allows the identification of local contrasts that closely resemble the projected structures and simulated ADF-STEM image of $Bi_2O_3$ along [100] (Fig. 1b). Specifically, the structural projection and simulated ADF-STEM image of ordered $Bi_2O_3$ feature a wave-shaped arrangement of Bi atomic columns from the [100] projection, while the experimental ADF-STEM image exhibits a similar contrast but with more lattice distortions (Fig. 1b). Accordingly, the FFT pattern of the experimental image exhibits rather diffuse spots compared with those in the FFT pattern of simulated image.

Followed by the secondary deposition of Pd, the clusters of $Bi_2O_3$ intergrown with Pd particles can be found according to the low-magnification ADF-STEM images (Supplementary Fig. 1). Besides, $Pd_{1.0}/Bi_2O_3/TiO_2$ exhibits no signals for Pd in the X-ray diffraction (XRD) patterns (Fig. 2a) while photo-deposited Pd/$TiO_2$ with an identical Pd loading (2.5 wt.%) shows a characteristic Pd(111) diffraction peak[29] at 40.1°, suggesting that pre-deposited $Bi_2O_3$ clusters assist in the high dispersion of Pd species in $Pd_{1.0}/Bi_2O_3/TiO_2$. According to the literatures[30,31], pre-deposited component with high work function can serve as a sink of photo-induced electrons, which preferentially reduce the second metal on the surface of the pre-deposited metal. By fixing the loading of Bi at 5.0 wt.%, we can also fabricate $Pd_{0.5}/Bi_2O_3/TiO_2$ (Supplementary Fig. 2), $Pd_{0.2}/Bi_2O_3/TiO_2$, and $Pd_{0.1}/Bi_2O_3/TiO_2$ with high dispersion of Pd ($Pd_x/Bi_2O_3/TiO_2$, $x$ represents Pd-to-Bi molar ratio). In contrast, when we increase Pd/Bi ratio

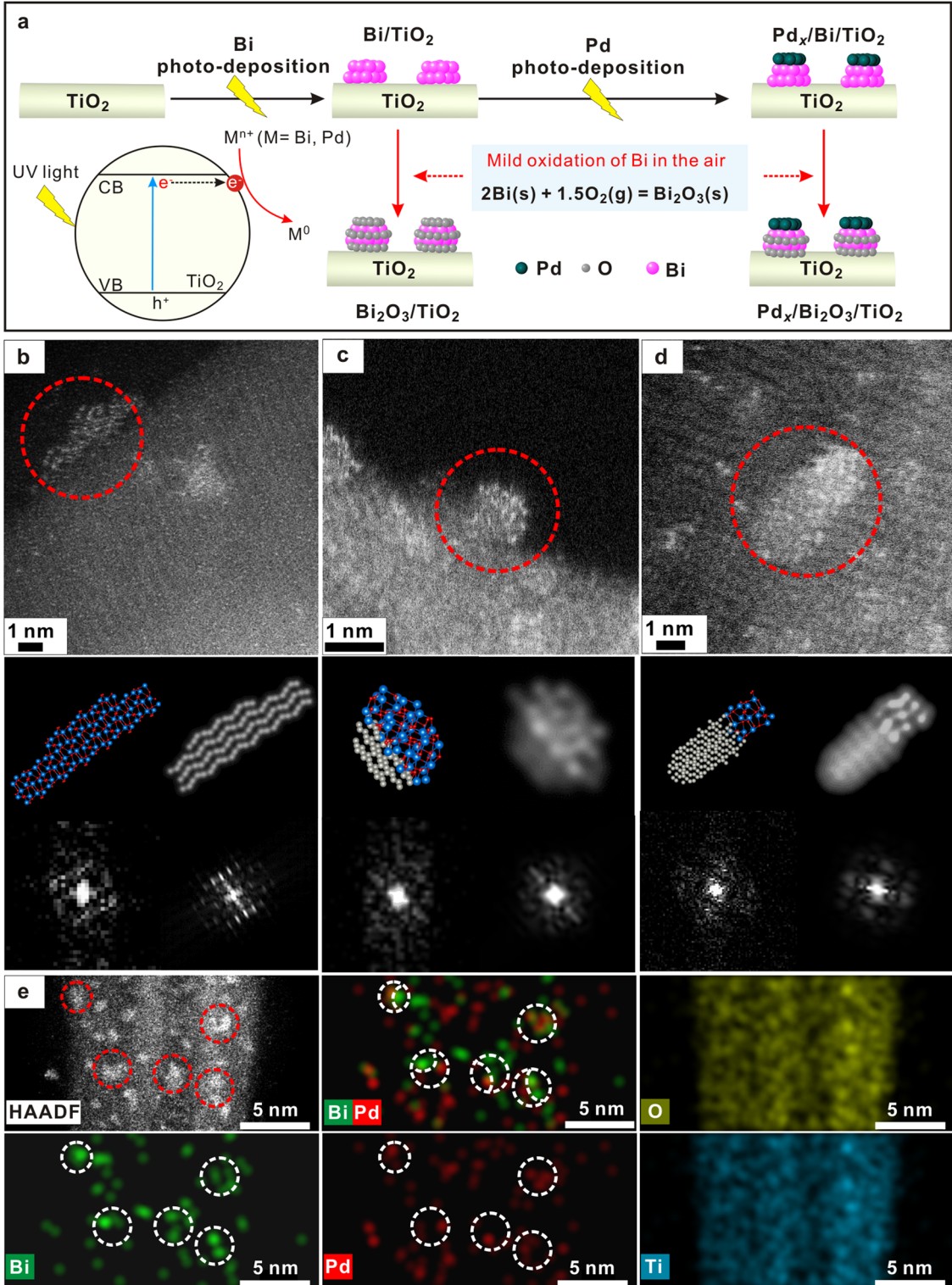

**Fig. 1 Microstructure of $Pd_{1.0}/Bi_2O_3/TiO_2$. a** Schematic illustration of the synthetic procedures. **b–d** STEM images of $Bi_2O_3/TiO_2$ (**b**) and $Pd_{1.0}/Bi_2O_3/TiO_2$ (**c, d**). The insets from upper to lower and left to right: HRSTEM images, projected structural models, simulated ADF-STEM images, FFTs from dashed circular regions in HRSTEM images, and FFTs from simulated ADF-STEM images, respectively. **e** Elemental mapping of $Pd_{1.0}/Bi_2O_3/TiO_2$.

to 3.0, the excess Pd also deposits directly on $TiO_2$. As a result, Pd nanoparticles are obtained (~6.7 nm, Supplementary Fig. 3). Taking all above results together, Pd species are most likely deposited on the existing $Bi_2O_3$ clusters although the discrimination between Pd and $Bi_2O_3$ components is not straightforward by

STEM imaging due to their small size and irradiation vulnerability. The nanoscopic elemental distribution can be mapped by a Super-X energy dispersive X-ray spectroscope (EDS) system with superior sensitivity. As shown in Fig. 1e and Supplementary Fig. 4, elemental mappings of Pd and Bi components suggest that

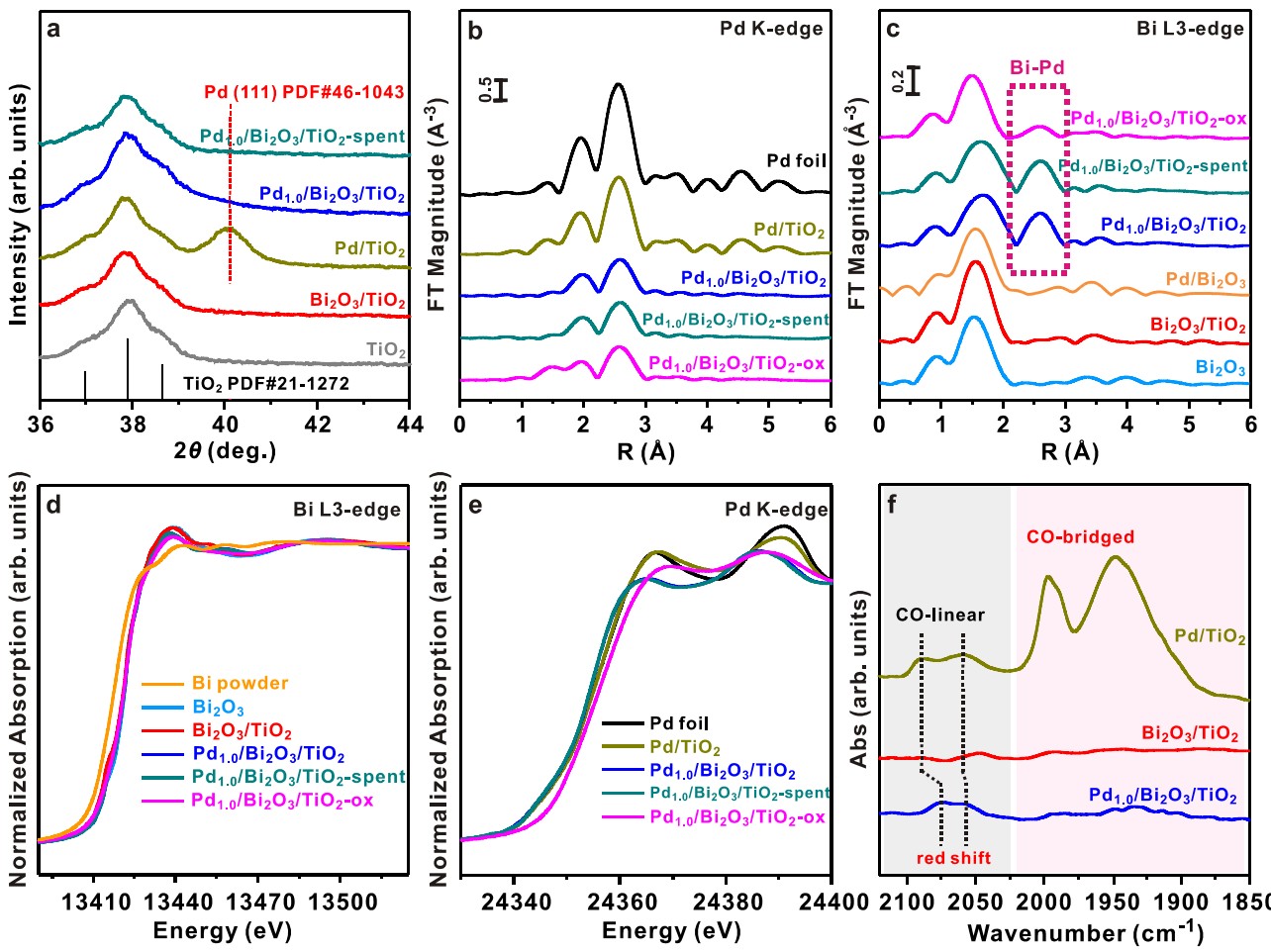

**Fig. 2 Characterization of $Pd_{1.0}/Bi_2O_3/TiO_2$. a** XRD patterns of $TiO_2$, $Pd/TiO_2$, $Bi_2O_3/TiO_2$, and $Pd_{1.0}/Bi_2O_3/TiO_2$; **b** Fourier transform spectra of Pd K-edge EXAFS for $Pd/TiO_2$, $Pd_{1.0}/Bi_2O_3/TiO_2$, and oxidized $Pd_{1.0}/Bi_2O_3/TiO_2$ ($Pd_{1.0}/Bi_2O_3/TiO_2$-ox); **c** Fourier transform spectra of Bi L3-edge EXAFS; **d** Bi L3-edge XANES spectra for $Bi_2O_3/TiO_2$, $Pd_{1.0}/Bi_2O_3/TiO_2$, and $Pd_{1.0}/Bi_2O_3/TiO_2$-ox. Bi and $Bi_2O_3$ powder were used as references. **e** Pd K-edge XANES spectra for $Pd/TiO_2$, $Pd_{1.0}/Bi_2O_3/TiO_2$, and $Pd_{1.0}/Bi_2O_3/TiO_2$-ox. Pd foil was used as a reference. **f** CO-adsorbed FT-IR spectra for various samples. Source data are provided in a Source data file.

their spatial distribution in most cases are correlated. In other words, these clusters are bicomponent with segregated Pd- and Bi-containing hemi-clusters, which unambiguously confirm the Pd-grafted $Bi_2O_3$ hybrid cluster structure. Careful inspection on several representative ADF-STEM images of $Pd_{1.0}/Bi_2O_3/TiO_2$ (Fig. 1c, d and Supplementary Fig. 5) allows the discrimination of Pd- and Bi-containing hemi-clusters from their distinct contrasts for adjacent clusters. The hemi-clusters featuring less bright contrast (marked by circle) are directly attached to brighter ones assigned to $Bi_2O_3$ hemi-clusters, further confirming that Pd is grafted onto the surface of $Bi_2O_3$ clusters. Notably, most such Pd clusters are identified to bond and hybridize with $Bi_2O_3$ clusters without any fixed orientation relationship or facet preference, probably due to the ultra-small size and large strain of the clusters. Two representative atomic-resolution ADF-STEM images of $Pd$-$Bi_2O_3$ nanoclusters with well-resolved Pd- and Bi-containing hemi-clusters are shown in Fig. 1c, d, of which the image contrasts and FFT patterns closely resemble the simulated ones of artificially constructed hybrid cluster models with different orientation relationships between two types of hemi-clusters made of Pd and $\alpha$-$Bi_2O_3$ structures, respectively. The above observations unambiguously validate the proposed $Pd$-$Bi_2O_3$ hybrid structural model. Quite rarely, such Pd clusters are observed to directly nucleate onto the $TiO_2$ substrate.

To confirm the coordination environment of Pd clusters, X-ray absorption fine structure (XAFS) of Pd K-edge and Bi L3-edge were performed, as shown in Fig. 2b–e. In the Fourier transformed extended X-ray absorption fine structure (FT-EXAFS) data of Bi L3-edge, it is important to note that a new coordination peak in R-space at about 2.6 Å is observed on $Pd_{1.0}/Bi_2O_3/TiO_2$ (Fig. 2c), but it is totally absent on other samples containing $Bi_2O_3$ (i.e., $Bi_2O_3$ and $Bi_2O_3/TiO_2$). Considering that the only difference between $Pd_{1.0}/Bi_2O_3/TiO_2$ and $Bi_2O_3/TiO_2$ is the secondary deposition of 2.3 wt.% Pd on $Bi_2O_3/TiO_2$, the peak at 2.6 Å should be contributed by the Bi–Pd interaction. Interestingly, the peak can be well fitted by a single Bi–Pd shell (Supplementary Fig. 6 and Supplementary Table 1), suggesting it is a characteristic peak of Bi–Pd coordination. To further prove this idea, the $Pd_{1.0}/Bi_2O_3/TiO_2$ is mildly treated under air at 150 °C (denoted as $Pd_{1.0}/Bi_2O_3/TiO_2$-ox) to oxidize Pd species while maintaining the structural integrity of $Bi_2O_3$ clusters[32]. In Fig. 2e, $Pd_{1.0}/Bi_2O_3/TiO_2$-ox shows clearly blue shift of the absorption edge position comparing with $Pd_{1.0}/Bi_2O_3/TiO_2$, indicating the oxidation of Pd after the treatment. Meanwhile, the peak at ~2.6 Å of Bi L3-edge EXAFS diminishes significantly upon oxidation of $Pd_{1.0}/Bi_2O_3/TiO_2$ (Fig. 2c). The fitting results reveal that the Bi–Pd CN is decreased from 2.9 for $Pd_{1.0}/Bi_2O_3/TiO_2$ to 1.1 for $Pd_{1.0}/Bi_2O_3/TiO_2$-ox (Supplementary Table 1). Accordingly, this experiment identifies that the peak at 2.6 Å is

from Bi–Pd bond and in turn proves the Pd–Bi pairs across the interfaces of Pd/$Bi_2O_3$ hybrid clusters observed by ADF-STEM in Fig. 1c, d. The Bi–Pd interaction eventually changes the coordination environment of Pd in $Pd_{1.0}$/$Bi_2O_3$/$TiO_2$. As compared to Pd foil and Pd/$TiO_2$, the significantly weaker and slightly broader coordination peak in the Pd $K$-edge EXAFS of $Pd_{1.0}$/$Bi_2O_3$/$TiO_2$ (Fig. 2b) implies a decreased CN of Pd–Pd pairs and distorted structure for the Pd clusters similarly as observed by STEM imaging[33]. The fitting of Pd $K$-edge EAXFS (Supplementary Fig. 7 and Supplementary Table 2) further confirms that the presence of Pd–Bi coordination (CN = 4.6) decreases the Pd–Pd CN from 10.0 for Pd/$TiO_2$ to 4.7 for $Pd_{1.0}$/$Bi_2O_3$/$TiO_2$, which is consistent with Pd cluster structure observed by ADF-STEM. The Pd–Bi bond length obtained from the fitting of Pd $K$ and Bi L3 edge is 2.79 ± 0.03 Å, suggesting the observed spatial correlation of Pd–Bi pairs arises from the direct bonding between Pd- and Bi-terminated clusters instead of the Pd–O–Bi moieties that attain a larger interatomic distance (~3.5 Å). Photo-deposition procedure is critical to ensure the formation of direct Pd–Bi bonding in Pd/$Bi_2O_3$ clusters as depicted in Fig. 1a. Theoretically, $Bi_2O_3$ favors an oxygen termination. Pd supported on $Bi_2O_3$ would be in direct contact with O rather than Bi. We prepared a Pd/$Bi_2O_3$ sample by directly depositing Pd onto commercial $Bi_2O_3$ support. Interestingly, no characteristic peak at ~2.6 Å was observed in the Bi L3 EXAFS of Pd/$Bi_2O_3$, excluding the presence of Pd–Bi bond (Fig. 2c). In this study, photo-deposition procedure ensures the Pd cluster deposited on reduced $Bi^0$ clusters and allows the formation of Pd–Bi bonding during the synthesis as depicted in Fig. 1a. The Pd–Bi bond preserves during the mild oxidation of Bi to $Bi_2O_3$ at room temperature (RT) that occurred after the deposition of Pd onto Bi/$TiO_2$, as evidenced by the characteristic Bi–Pd peak in Bi L3 EXAFS (Fig. 2c). More interestingly, 36% of this peak preserves even after oxidation in air at a higher temperature of 150 °C ($Pd_{1.0}$/$Bi_2O_3$/$TiO_2$-ox). These results clearly indicate the good stability of Pd–Bi bond in $Pd_{1.0}$/$Bi_2O_3$/ $TiO_2$ and suggest that Pd supported on Bi-terminated $Bi_2O_3$ as the structural model for $Pd_{1.0}$/$Bi_2O_3$/$TiO_2$ in the hydrogenation reaction conditions. Taken together, the above-mentioned results unambiguously confirmed that the Pd species are predominantly in the form of nanometer-sized Pd clusters embedded in the Pd/ $Bi_2O_3$ hybrid clusters through chemical adhesion.

In order to identify the electronic structure of the as-synthesized Pd/$Bi_2O_3$ hybrid clusters in reaction conditions, in situ X-ray photoelectron spectra (XPS) was collected at 100 °C under $H_2$ atmosphere (Supplementary Fig. 8). As shown in Supplementary Fig. 8b, there are symmetric Bi 4$f$ peaks at 158.8/ 164.1 eV, confirming that the majority of Bi are in the form of $Bi_2O_3$[34]. Both $Bi_2O_3$/$TiO_2$ and $Pd_{1.0}$/$Bi_2O_3$/$TiO_2$ exhibit similar Bi L3-edge structures with $Bi_2O_3$ by inspecting the X-ray absorption near edge structure (XANES), indicating similar valence states among them (Fig. 2d). It is frequently reported that intermetallic compounds (IMC) could be possibly formed by a so-called reactive metal–support interaction[35]. However, according to the density functional theory (DFT) calculations (Supplementary Table 3), the Pd–Bi distance in PdBi IMC ranges from 2.85 to 3.03 Å, which is obviously larger than that of $Pd_{1.0}$Bi/$TiO_2$ (2.79 Å, Supplementary Table 2). This result excludes PdBi IMC as the main phase of $Pd_{1.0}$/$Bi_2O_3$/$TiO_2$. To better illustrate the difference between Pd–$Bi_2O_3$ nanoclusters and PdBi intermetallic compounds, we synthesized a PdBi IMC (denoted as PdBi/$TiO_2$) for comparison (Supplementary Fig. 9). XRD patterns (Supplementary Fig. 9a) of PdBi/$TiO_2$ exhibit characteristic peaks at 39.9 and 42.8°, corresponding to (102) and (110) of hexagonal PdBi IMC (sobolevskite, $P6_3$/mmc(194), $a = b = 4.22$ Å, $c = 5.709$ Å, $\alpha = \beta = 90°$, and $\gamma = 120°$). Pd–M (M = Pd or Bi) coordination peaks in $R$ space of PdBi/$TiO_2$ obviously shift (~0.06 Å) from that

of $Pd_{1.0}$/$Bi_2O_3$/$TiO_2$, confirming that PdBi IMC have longer Pd–Bi distances than Pd–$Bi_2O_3$ nanoclusters (Supplementary Fig. 9b). More importantly, a massive $Bi^0$ peak at 156.6/162.0 eV (Supplementary Fig. 9d) is observed in the Bi 4$f$ of PdBi/$TiO_2$ but is absent in that of $Pd_{1.0}$/$Bi_2O_3$/$TiO_2$ (Supplementary Fig. 8b). These results clearly indicate that $Pd_{1.0}$/$Bi_2O_3$/$TiO_2$ is composed of Pd–$Bi_2O_3$ hybrid clusters rather than PdBi IMC. It is reasonable because the reaction temperature in this study is too low to reduce $Bi_2O_3$ to Bi, which is a prerequisite for the formation of PdBi IMC. On the other side, the Pd 3$d$ patterns of Pd/$TiO_2$ and $Pd_{1.0}$/$Bi_2O_3$/$TiO_2$ exhibit a slightly asymmetric lineshape (Supplementary Fig. 8a), which is most likely due to the many-body screening response of conduction electrons to the photoemission of a core electron[36]. The predominant Pd 3$d$ signals of Pd/$TiO_2$ and $Pd_{1.0}$/$Bi_2O_3$/$TiO_2$ locate at ~334.9/ 340.0 eV, which are characteristic of $Pd^0$. It is important to note that, due to the differences in the extra-atomic relaxation of metal particles of different sizes, decreasing the particle size of Pd generally upshifts Pd 3$d$ to higher binding energy (BE)[37]. In this study, the Pd 3$d$ and 3$p$ of $Pd_{1.0}$/$Bi_2O_3$/$TiO_2$ downshift slightly to lower BE although the particle size of $Pd_{1.0}$/$Bi_2O_3$/$TiO_2$ is significantly smaller than that of Pd/$TiO_2$ (Supplementary Figs. 8a, c). These results, opposite to the particle-size-induced BE shift, indicate a charge transfer from Bi to Pd. Similar result was also reported in Au/$TiO_2$ system[38].

The more important feature associated with the electronic structure of Pd clusters is observed in the Pd $K$-edge and L3-edge structures as shown in Fig. 2e and Supplementary Fig. 10. Specifically, the Pd $K$-edge XANES profile of Pd/$TiO_2$ is very similar to that of Pd foil. In contrast, $Pd_{1.0}$/$Bi_2O_3$/$TiO_2$ shows a marked red-shift of the absorption edge energy and decrease in the white line intensity. This indicates the electron-richness of Pd atoms in $Pd_{1.0}$Bi/$TiO_2$ compared to those in Pd foil and Pd/$TiO_2$, which most likely arises from the $Bi_2O_3$-to-Pd intra-cluster electron transfer. Moreover, $Pd_{1.0}$/$Bi_2O_3$/$TiO_2$ shows a much weaker Pd L3 white line intensity at ~3173 eV than Pd/$TiO_2$ (Supplementary Fig. 10), which indicates an enhanced $d$-band filling. Such phenomenon is similarly predicted by the Bader charge analysis over $Pd_8$ clusters supported by the $Bi_2O_3$ cluster as shown in Supplementary Fig. 11. It is quite surprising to observe such an enhanced $d$-band filling and associated negative shift of $d$-band center away from the Fermi level for nanometer-sized Pd clusters, because the strong "size effect" of most supported metals usually leads to a decreased $d$-band filling and thus positive shift of $d$-band center toward Fermi level for smaller nanoparticles[39]. Actually, this has become a limiting factor for applying supported metal nanoparticles in the selective acetylene hydrogenation reaction due to the strong adsorption of ethylene molecules on electron-deficient metals and results in over-hydrogenation. The metal termination of $Bi_2O_3$ observed here at the Pd–Bi interface of the hybrid cluster could lead to a strong downshift of its surface conduction band. Similar results were also reported in metal–metal interface of Pd/MgO, Pd/ZnO, Ru/ MgO, and Au/$TiO_2$[38,40–42]. Electrons transferred from terminated magnesium or zinc to adsorbed Pd can also result in negatively charged Pd. This charge transfer was ascribed to the band filling modification and the orbital hybridization between substrate and metal atoms[40]. Specifically, for Pd deposited on magnesium termination, the surface Mg conduction band is shifted toward higher energy and is emptied, while the Pd $d$ band is shifted in the opposite direction and becomes filled. Similar band filling modification and orbital hybridization might also apply to Pd–$Bi_2O_3$ system. This allows the $Bi_2O_3$-to-Pd intra-cluster electron transfer through direct Pd–Bi bonding across the interface, which leads to a greater filling of high-lying $d$ bands in Pd clusters and largely circumvented over-hydrogenation

problem. It is important to note that the charge transfer between Pd and Bi is localized in the Pd–Bi interface. Considering the relatively higher concentration of $Bi_2O_3$ comparing with Pd, the signal of electron-deficient Bi is likely overwhelmed by the signal of unaffected $Bi_2O_3$ and is therefore not observed by XPS and XANES.

The unique atomic and electronic structures of heterografted Pd cluster further result in its largely modulated gas adsorption behaviors compared with other supported Pd metals, which can be investigated by the in situ diffuse reflectance infrared Fourier transform spectroscopy using CO as the probe molecule. As shown in Fig. 2f, signals at 2089 and 2059 cm$^{-1}$ for linear-bonded CO[43] over $Pd/TiO_2$ redshift to 2076 and 2055 cm$^{-1}$ for those over $Pd_{1.0}/Bi_2O_3/TiO_2$, suggesting a strengthened π-back donation of metal $d$-electrons to π* orbitals of CO over $Pd_{1.0}/Bi_2O_3/TiO_2$ and a downshift of $d$-band center[44]. This result is well consistent with the XPS and XANES results. It is noteworthy that signals for bridge-bonded CO (1996 and 1948 cm$^{-1}$) are observed over $Pd/TiO_2$ but invisible over $Pd_{1.0}/Bi_2O_3/TiO_2$, likely arising from combined size and electronic effects. Specifically, the downshifted $d$-band center weakens the adsorption strength of CO on Pd[44]. More importantly, the small size and large structural distortion of Pd clusters disfavor the bridge adsorption mode of CO molecules due to the low average CN and broad distance distribution of Pd–Pd pairs.

**Catalytic performance in acetylene hydrogenation**. With the unique hybrid cluster structure and $Bi_2O_3$-to-Pd intra-cluster electron transfer, $Pd_{1.0}/Bi_2O_3/TiO_2$ readily serves as a model catalyst to demonstrate the essential catalytic role of a nanometer metal/oxide interface. The catalytic properties were evaluated in selective hydrogenation of acetylene with excess ethylene, mimicking the front-end condition. In this condition, the thermodynamically favored over-hydrogenation of ethylene with the large excess hydrogen generally leads to an unsatisfied $C_2H_4$ selectivity at high $C_2H_2$ conversion, accompanied by a thermal runaway. To overcome this problem, a small amount of CO is usually added in the feed stream to reduce the reaction rate so as to improve the $C_2H_4$ selectivity[10]. At very low CO levels, high $C_2H_4$ selectivity and $C_2H_2$ conversion are difficult to achieve simultaneously. In this study, no CO is added in the feed stream. Both $Pd/TiO_2$ and well-established $Pd_1Ag_3/Al_2O_3$ catalysts[30,45] were evaluated for comparison with $Pd_{1.0}/Bi_2O_3/TiO_2$. The composition and synthesis procedure of the $PdAg_3/Al_2O_3$ catalyst is the same as OleMax@251, a widely used industrial catalyst for acetylene hydrogenation[45]. The carbon balances are all >99%. Negligible oligomers were formed during the hydrogenation process, likely due to the short contact time and high concentration of hydrogen. According to the literatures, the large excess of hydrogen would change the adsorption modes of $C_2H_2$ and $C_2H_4$ from C-CH$_2$ vinylidene and C-CH$_3$ ethylidyne to weak π-bonded $C_2H_2$ and $C_2H_4$[46,47]. As a result, the possible reaction between vinylidene and acetylene to form $C_4$ species as well as the hydrocarbon isomerization and decomposition are suppressed.

Figure 3a plots the $C_2H_4$ selectivity as a function of $C_2H_2$ conversion on various catalysts. Consistent with the literature results[30], the $C_2H_4$ selectivity drops rapidly on well-established $Pd_1Ag_3/Al_2O_3$ catalyst once the $C_2H_2$ conversion exceeds 40%. On the contrary, $Pd_{1.0}/Bi_2O_3/TiO_2$ and $Pd_{0.2}/Bi_2O_3/TiO_2$ catalysts preserve very high selectivity toward $C_2H_4$ at much higher $C_2H_2$ conversions. Figure 3b compares the $C_2H_4$ selectivity at 95% $C_2H_2$ conversion. Interestingly, all catalysts except $Pd_{1.0}/Bi_2O_3/TiO_2$ and $Pd_{0.2}/Bi_2O_3/TiO_2$ exhibit negative selectivity toward $C_2H_4$ due to the over-hydrogenation of ethylene to ethane. Specifically, $Pd_{1.0}/Bi_2O_3/TiO_2$ exhibits much higher selectivity

than 2.3 wt.% $Pd/TiO_2$ regardless of the same Pd loading. In addition, we also synthesized $Bi_x/Pd/TiO_2$ ($x = 0.5$, 1, $x$ is the molar ratio of Bi to Pd) by photo-depositing Bi onto 2.3 wt% $Pd/TiO_2$. Interestingly, both $Bi_{0.5}/Pd/TiO_2$ and $Bi_{1.0}/Pd/TiO_2$ exhibit 100% conversion of $C_2H_2$ and negative selectivity toward ethylene (−123% for $Bi_{0.5}/Pd/TiO_2$ and −143% for $Bi_{0.5}/Pd/TiO_2$) at RT. These results suggest that a simple site blocking mechanism cannot explain the beneficial effect of Bi in this study. $PdBi/TiO_2$ composed of PdBi IMC (Supplementary Fig. 9) shows 100% conversion of $C_2H_2$ and negative selectivity toward ethylene (−1319%) at RT. The strong exothermic effect of unselective acetylene hydrogenation eventually leads to a runaway temperature up to 58.5 °C. These results exclude PdBi IMC as the active site for $Pd_{1.0}/Bi_2O_3/TiO_2$ and further indicates the critical role of the nanometer $Pd/Bi_2O_3$ interface in the catalytic selectivity of Pd. It is important to highlight that 91% selectivity of $C_2H_4$ with 90% conversion of $C_2H_2$ is achieved by $Pd_{1.0}/Bi_2O_3/TiO_2$ at a temperature as low as 44 °C (Fig. 3c). Such an excellent low-temperature performance of acetylene hydrogenation has never been reported previously, suggesting the unique structure and catalytic properties of $Pd_{1.0}/Bi_2O_3/TiO_2$. Moreover, the $C_2H_2$ conversion and $C_2H_4$ selectivity remain almost constant over 24 h operating at 40 °C (Fig. 3d). XRD and XAFS also confirm that $Pd/Bi_2O_3$ hybrid cluster structure is still maintained (Fig. 2), demonstrating a good long-term stability of $Pd_{1.0}/Bi_2O_3/TiO_2$.

The variation of Pd-to-Bi ratio leads to an evolution of $Pd/Bi_2O_3$ hybrid clusters and major alteration of their catalytic performances in selective acetylene hydrogenation. Usually, the high Pd-to-Bi ratio results in the formation of Pd nanoparticles along with decreased $Pd/Bi_2O_3$ clusters, while low Pd-to-Bi ratio results in Pd single atoms. Here the reaction temperature ($T_{90}$) and $C_2H_4$ selectivity at 90% conversion of acetylene are utilized to evaluate the catalytic performance of the samples. As shown in Fig. 3c, $T_{90}$ increases along with the decrease of Pd-to-Bi ratio, implying that small Pd size is unfavorable to hydrogenation. Under an extreme condition when Pd is atomically dispersed (Pd-to-Bi ratio ≤0.1), a very high $T_{90}$ (>90 °C) will be obtained, likely due to its poor ability of hydrogen activation[21]. In contrast, the moderate size of Pd in the hybrid clusters allows the efficient activation of hydrogen at much lower temperatures. All these catalysts exhibit quite high $C_2H_4$ selectivity. The $Pd_{3.0}/Bi_2O_3/TiO_2$ with an increased Pd-to-Bi ratio generates Pd nanoparticles and can even convert >90% of acetylene at RT. However, it suffers from the negative $C_2H_4$ selectivity at 90% conversion of acetylene. The above results experimentally confirm the trade-off between the conversion of acetylene and the selectivity of $C_2H_4$ upon the size effect of Pd, while an optimal catalytic performance is achieved over the Pd clusters stabilized by a nanometer metal–oxide interface.

The origin of such inherent trade-off in the catalytic performance of Pd lies in two aspects: the hydrogen activation and ethylene adsorption. It is generally accepted that the facile dissociative activation of hydrogen on Pd nanoparticles produces too much active H species, which migrate into the Pd lattice and generate β-hydride phase that leads to over-hydrogenation of ethylene[43]. Accordingly, a negative peak (55–75 °C, Fig. 3e) characteristic for the decomposition of β-hydride phase is observed from the temperature programmed reduction (TPR) profile of $Pd/TiO_2$, which is, however, not observed from that of $Pd_{1.0}/Bi_2O_3/TiO_2$. These results are consistent with literatures that report high-coordinated Pd ensembles (the case of $Pd/TiO_2$) as active sites for the formation of β-hydride[43]. As for $Pd_{1.0}/Bi_2O_3/TiO_2$, the $Pd/Bi_2O_3$ hybrid clusters feature a small Pd–Pd CN and nanometer Pd–Bi interface, which prevent the formation of β-hydride. The lack of β-hydride suppresses the over-hydrogenation of ethylene to ethane and therefore contribute to

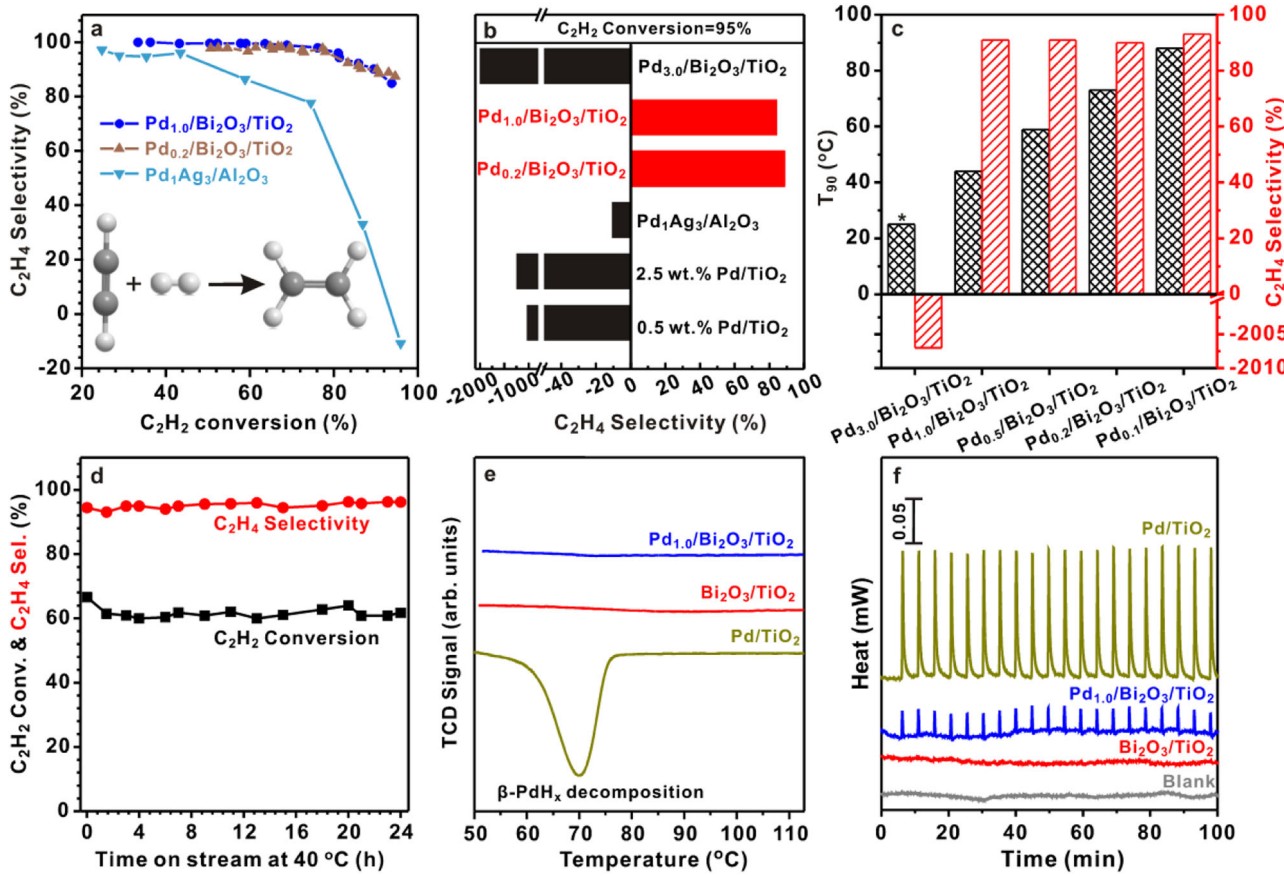

**Fig. 3 Catalytic performances of $Pd_{1.0}/Bi_2O_3/TiO_2$ in acetylene hydrogenation. a** Selectivity as a function of acetylene conversion over $Pd_{1.0}/Bi_2O_3/TiO_2$, $Pd_{0.2}/Bi_2O_3/TiO_2$, and $PdAg_3/Al_2O_3$. **b** The selectivity to $C_2H_4$ for 95% acetylene conversion over different catalysts. **c** Reaction temperature ($T_{90}$) and $C_2H_4$ selectivity for 90% acetylene conversion. *For $Pd_{3.0}/Bi_2O_3/TiO_2$, hydrogen dissociation easily takes place at room temperature. The strong exothermic effect of unselective acetylene hydrogenation eventually leads to a runaway temperature up to 63.5 °C. **d** $C_2H_2$ conversion and $C_2H_4$ selectivity with time on stream over $Pd_{1.0}/Bi_2O_3/TiO_2$ at 40 °C. **e** $H_2$-TPR profiles for $Pd/TiO_2$, $Bi_2O_3/TiO_2$, and $Pd_{1.0}/Bi_2O_3/TiO_2$. **f** Microcalorimetric studies of $C_2H_4$ pulse adsorption over $Pd/TiO_2$ and $Pd_{1.0}/Bi_2O_3/TiO_2$. Source data are provided in a Source data file.

the high $C_2H_4$ selectivity[48]. It is important to highlight that the reduced size of Pd cluster does not compromise the hydrogen activation and thus the catalytic activity. As a result, a low $T_{90}$ could be achieved on $Pd_{1.0}/Bi_2O_3/TiO_2$ (Fig. 3c).

In addition to the formation of β-hydride, the adsorption behavior of ethylene is also strongly influenced by the Pd structures, which leads to different microcalorimetric profiles[49]. Figure 3f shows the heat flow during the $C_2H_4$ pulse adsorption process as a function of time for $Pd_{1.0}/Bi_2O_3/TiO_2$, $Pd/TiO_2$, and $Bi_2O_3/TiO_2$ at 40 °C. Obvious heat flow signals are observed on $Pd/TiO_2$ and $Pd_{1.0}/Bi_2O_3/TiO_2$ but are absent on $Bi_2O_3/TiO_2$ and blank test, demonstrating that $C_2H_4$ is adsorbed on Pd instead of on $Bi_2O_3$. It is interesting to note that the amplitude of the heat flow signal of $Pd_{1.0}/Bi_2O_3/TiO_2$ is much smaller than that of $Pd/TiO_2$. The calculated adsorption heat of $Pd_{1.0}/Bi_2O_3/TiO_2$ (~5.9 kJ mol$^{-1}$) is significantly lower than that of $Pd/TiO_2$ (~234.5 kJ mol$^{-1}$), clearly indicating a much weaker ethylene adsorption on $Pd_{1.0}/Bi_2O_3/TiO_2$. These observations can be attributed to the unique geometric and electronic structure of hybrid cluster, similar to the results reported in the alloying of Pd[50,51]. The low Pd–Pd coordination and the $Bi_2O_3$-to-Pd intra-cluster electron transfer likely change the adsorption configuration of $C_2H_4$ from stable ethylidyne to weak π-bonded $C_2H_4$ and promote the desorption of ethylene as the desired product. To confirm this hypothesis, we further performed the temperature programmed desorption of ethylene by monitoring the mass signal of $m/e = 27$ (Supplementary Fig. 12). According to the

literature, the peak at ~65 °C could be assigned to weak π-bonded ethylene species, which readily desorb without decomposition[52]. The peak centered at ~115 °C originates from di-σ-bonded ethylene, which undergoes decomposition followed by the recombination of surface hydrocarbon species and hydrogen to produce ethylene and ethane[52]. Compared with $Pd/TiO_2$, $Pd_{1.0}/Bi_2O_3/TiO_2$ exhibits a much weaker peak at ~115 °C but a significantly larger peak at 65 °C. These results confirm that the adsorption configuration of $C_2H_4$ is changed from the strong σ-bonding for $Pd/TiO_2$ to weak π-bonding for $Pd_{1.0}/Bi_2O_3/TiO_2$.

**Reaction mechanism investigated by DFT calculations.** DFT calculations were performed to further provide insights into the molecular-level mechanisms of acetylene hydrogenation on experimental $Pd_{1.0}/Bi_2O_3/TiO_2$ catalyst. Model of $Bi_2O_3$-supported $Pd_8$ cluster catalyst was built according to the experimental characterization results and the structure of which is shown in Fig. 4a (please see the details of model development in Supplementary Information). In this model, the size of Pd cluster on $Bi_2O_3(100)$ is around 1.6 nm × 1.5 nm. In addition, this Pd cluster shows an average Pd–Pd CN of 4.0, which is close to the experimental values measured, i.e., 4.7 ± 0.5 (Supplementary Table 2). The Pd–Bi pair distribution function of the $Pd_8$ cluster structure is shown in Supplementary Fig. 13. In this figure, the dominant peak appears at ~2.75 Å, which is smaller than that in

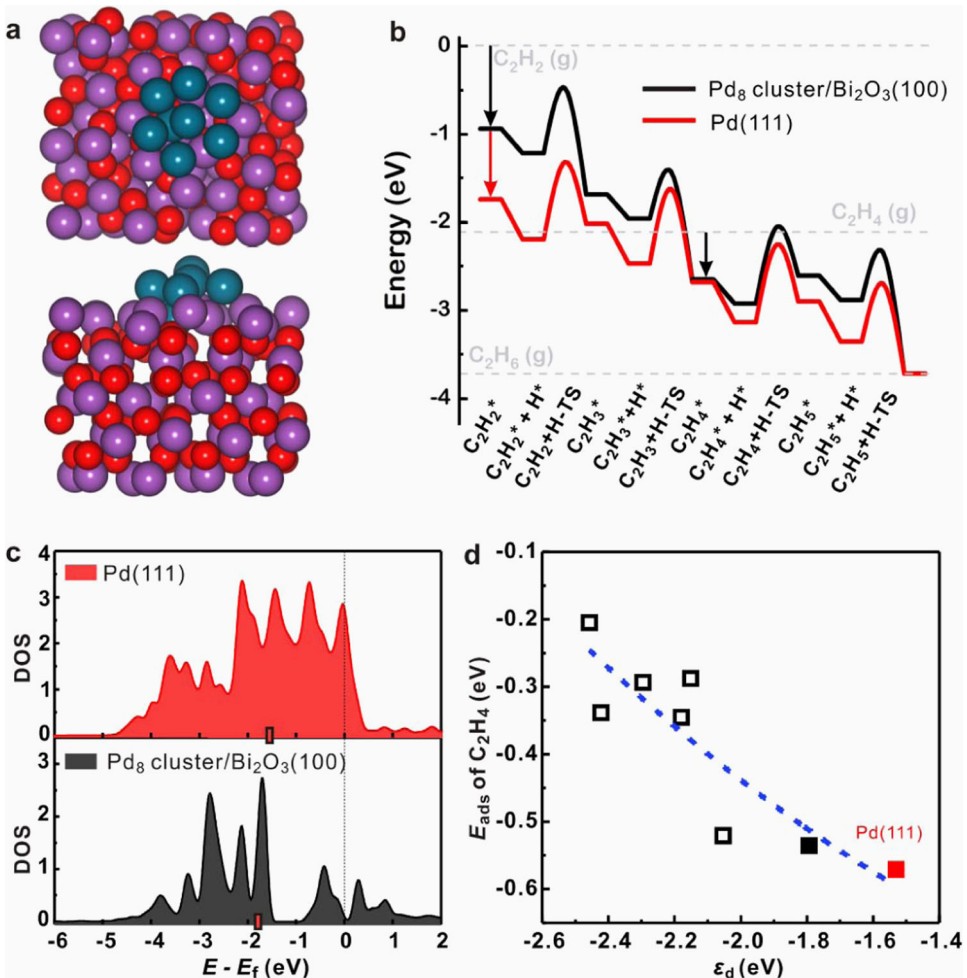

**Fig. 4 Reaction mechanism revealed by DFT calculations. a** Optimized Pd cluster structure for DFT calculation (Pd: cyan, Bi: purple, O: red) and **b** energy profile of acetylene hydrogenation to ethane on Pd(111) and $Pd_8$ cluster supported on $Bi_2O_3$(100). **c** DOS projected onto *d* electrons over Pd atom of Pd(111) and $Pd_8$ cluster structures. A surface Pd atom of Pd(111), and the most active Pd atom of $Pd_8$ cluster structure (on which $C_2H_4$ adsorbs most strongly) are chosen to plot the DOS. The position of *d*-band center ($\varepsilon_d$) is highlighted with a red bar. **d** $E_{ads}$ of $C_2H_4$ as a function of $\varepsilon_d$ over different Pd atom on Pd cluster surface (black squares). The most stable adsorption configuration is shown as solid square, while the other less stable adsorption structures are denoted by hollow squares. A surface Pd atom of Pd(111) is also shown as red solid square for comparison. The blue fitted line is a guide for the eyes. It shows that a more negative $\varepsilon_d$ corresponds to a more positive $E_{ads}$ of $C_2H_4$. Source data are provided in a Source data file.

the PdBi intermetallic model (~2.91 Å, Supplementary Table 3). In addition, Bader charge analysis suggests that Pd atoms in PdBi IMC (average charge of −0.36 e) model are more electron-rich than those in Pd-$Bi_2O_3$ hybrid clusters model (−0.21 e). These results are well consistent with Pd *K*-edge XANES and Pd 3*d* XPS shown in Supplementary Fig. 9, which therefore validate the reliability of the $Pd_8$ cluster model. The possibility of hydride formation over the Pd cluster was studied, and it was found that hydrogen atoms prefer to adsorb at surface Pd sites after optimization (Supplementary Fig. 14), suggesting that formation of Pd hydride from this cluster is difficult, which again agrees well with the results shown in Fig. 3e.

It was reported that C-$CH_2$ vinylidene and C-$CH_3$ ethylidyne are energetic stable and important spectator species in acetylene hydrogenation[53–55]. However, in the front-end condition where $H_2$ is in large excess, vinylidene and ethylidyne are readily hydrogenated and are insignificant at steady-state conditions[10]. Besides, the adsorption configurations of $C_2H_2$ and $C_2H_4$ also depend on the catalyst structure. For the Pd cluster structure studied in this work, it is found that the hydrogenation of CH≡CH to $CHCH_2$ has the lowest activation energy ($E_a$) of 0.74 eV compared with the dehydrogenation or hydrogen shift of

CH≡CH (Supplementary Table 4). This strongly suggests that formation of $CCH_2$ would be unfavorable on the Pd cluster structure. Similarly, the hydrogenation of $CHCH_2$ to $CH_2CH_2$ has the lowest $E_a$ among all reactions starting from $CHCH_2$, again indicating that the spectator species $CCH_3$ is hard to form on Pd cluster model (please see Supplementary Information for details). To this end, the effects of these species are not discussed here. The energy profile of acetylene hydrogenation to ethane over this Pd cluster is shown in Fig. 4b, and the corresponding optimized configurations of surface intermediates and transition states are shown in Supplementary Fig. 15. Meanwhile, the energy profile on Pd(111) representing Pd foil is also shown in Fig. 4b for comparison. The calculated barriers are generally consistent with the reported values (Supplementary Table 5), demonstrating that our calculated results are reliable. In addition, we further calculated the vibrational frequency based on the transition state structures on Pd(111). All the transition states were characterized to possess only one imaginary frequency along the bond formation of hydrogenation reactions. These results further demonstrate the reliability of the constrained minimization method that we used in this work. As can be seen from Fig. 4b, the adsorption of $C_2H_2$ over Pd(111) and $Pd_8$ cluster is

exothermic, and the transition state energies of $C_2H_2$ hydrogenation to $C_2H_4$ on both surfaces are below the energy of gaseous $C_2H_2$, suggesting that the hydrogenation processes should be facile. The semi-hydrogenation product $C_2H_4$ would either desorb from the surface or undergo further hydrogenation to ethane ($C_2H_6$). In Fig. 4b, the transition state energy of $C_2H_4$ hydrogenation on $Pd_8$ cluster is above the gaseous $C_2H_4$ energy, suggesting that desorption of $C_2H_4$ from the $Pd_8$ cluster may be favored compared with its further hydrogenation to $C_2H_5$. Herein the difference between the barriers for further hydrogenation and desorption of ethylene can be used to estimate the possibility of selective $C_2H_4$ formation[17,56–60]. Within this framework, a more positive value of $E_{a,hydro} - |E_{ads,C2H4}|$, where $E_{a,hydro}$ is the effective hydrogenation barrier of $C_2H_4$ to $C_2H_6$ and $|E_{ads,C2H4}|$ is the absolute value of $C_2H_4$ adsorption energy, corresponds to higher $C_2H_4$ selectivity. The calculated values of $E_{a,hydro} - |E_{ads,C2H4}|$ over $Pd_8$ cluster and Pd(111) are 0.53 and 0.31 eV, respectively, demonstrating higher $C_2H_4$ selectivity over the $Pd_8$ cluster than Pd(111), which is consistent with the experimental results.

We further carried out electronic structure analysis to understand the weaker adsorption of reaction intermediates over the Pd cluster than Pd(111). In Fig. 4c, we plotted the density of states projected onto $d$-electrons of the surface Pd atom where $C_2H_4$ adsorbs over Pd(111) and Pd cluster, and the $d$-band center ($\varepsilon_d$) were calculated to be −1.53 and −1.79 eV, respectively. According to the $d$-band center theory[61], the more negative $\varepsilon_d$ indicates weaker binding to adsorbates, which is in agreement with the trend of adsorption energies calculated for the reaction intermediates over Pd(111) and Pd cluster. In addition, the $\varepsilon_d$ of each Pd atom on Pd cluster was found to be correlated with $E_{ads}$ of $C_2H_4$ as shown in Fig. 4d, and a more negative $\varepsilon_d$ generally corresponds to weaker adsorption of $C_2H_4$. The most stable adsorption of ethylene, corresponding to the black solid square in Fig. 4d and *$C_2H_4$ in Fig. 4b, gives rise to the $E_{ads}$ of −0.54 eV, which is slightly weaker than that over Pd(111), i.e., −0.57 eV. However, one can also see that $C_2H_4$ adsorbs much weaker at other Pd sites over the $Pd_8$ cluster (shown as black hollow squares), therefore $Pd_8$ cluster shows weaker adsorption on average compared with those sites over Pd(111) where all the surface sites are identical. $C_2H_4$ would be more likely to desorb from the Pd cluster, resulting in the high $C_2H_4$ selectivity observed.

Our findings demonstrate the essential catalytic roles of a nanometer metal/oxide interface in selective hydrogenation of acetylene. Pd–$Bi_2O_3$ hybrid clusters feature a small Pd–Pd CN and intra-cluster electron transfer, which enables a weak $C_2H_4$ adsorption without compromising the $H_2$ activation activity. The superior low-temperature catalytic performance of Pd–$Bi_2O_3$ nanocluster ensembles over Pd single atom and nanoparticles might open new opportunities for fundamental research of hybrid nanoclusters. Besides, the demonstrated stepwise photochemical strategy also provides a new path for fabricating hybrid nanoclusters and nanometer metal/oxide interface.

## Methods

**Synthesis of $Pd_x$/$Bi_2O_3$/$TiO_2$.** $Pd_x$/$Bi_2O_3$/$TiO_2$ ($x$ is the nominal molar ratio of Pd to Bi) catalysts were prepared by a two-step photo-deposition method using a high-pressure Xe lamp (300 W) as the light source. Typically, 100 mg of $TiO_2$ and 11.6 mg of $Bi(NO_3)_3$·$5H_2O$ (Sinopharm chemicals, 99%) were dispersed in 4 mL of ethylene glycol in a Pyrex glass reactor. Prior to ultraviolet irradiation for 1 h, the suspension was bubbled with Ar for 30 min to eliminate dissolved $O_2$. Subsequently, 8 mL of $PdCl_2$ aqueous solution (the concentration is determined by $x$) was added into the suspension. After irradiation in Ar for another 1 h, the precipitates were collected by centrifugation, washed twice by water and ethanol, and then dried in an oven at 40 °C. When the sample was exposed in the air, Bi was oxidized to $Bi_2O_3$ spontaneously. The catalyst was activated by $H_2$ at 100 °C for 1 h and then cooled to RT in $N_2$ prior to the catalytic reaction and characterizations.

The synthetic procedures of $Bi_2O_3$/$TiO_2$, Pd/$TiO_2$, $Pd_{1.0}$/$Bi_2O_3$/$TiO_2$-ox, PdBi/$TiO_2$, and Pd/$Bi_2O_3$ are presented in Supplementary Information.

**Characterization.** Powder XRD patterns were recorded on a Rigaku Ultimate IV diffractometer using Cu Kα radiation operated at 40 mA and 40 kV (scan rate: 5º $min^{-1}$). XPS measurements were performed in a VG Scientific ESCALAB Mark II spectrometer. All BEs were referenced to the C 1s peak at 284.8 eV of the surface adventitious carbon to correct the shift caused by charge effect. In situ near ambient pressure XPS (NAP-XPS) was conducted using a Specs NAP-XPS system with a PHOIBOS NAP hemispherical energy analyzer in Vacuum Interconnected Nanotech Workstation, Suzhou Institute of Nano-Tech and Nano-Bionics. The NAP operation was conducted under 1 mbar $H_2$ atmosphere. The actual loading of Pd and Bi were analyzed by ICP-AES using a Profile Spec ICP-AES spectrometer (Leeman, USA). Structural characterizations and chemical composition distribution were determined by a probe Cs-corrected electron microscope (FEI Titan) equipped with a Super-X EDS. Microcalorimetric measurements of $C_2H_4$ adsorption were carried out on a home-designed adsorption microcalorimetry system consisting of a chemisorption apparatus (Micromeritics Autochem II 2920) and a microcalorimeter (Setaram Sensys EVO 600).

The XAFS spectra at Pd K-edge and Bi L3-edge of the samples were measured at beamline 14W of Shanghai Synchrotron Radiation Facility in China[62]. The output beam was filtered by Si(311) monochromator. Pd foil was used to calibrate the energy.

In situ Fourier transform infrared (FT-IR) adsorption spectroscopy of CO experiments were recorded on a Nicolet iS50 instrument. Prior to CO chemisorption, the catalysts were activated by $H_2$ at 100 °C for 1 h and then cooled down to RT in Ar. The FT-IR spectrum of Ar at RT was taken as the background spectrum and subtracted automatically from subsequent spectra. Then the catalysts were exposed to a CO flow for 10 min and degassed by Ar for 10 min to desorb the physical adsorbed CO, and IR spectra were recorded.

TPR measurements were conducted on Micromeritics ChemiSorb 2920, equipped with a thermal conductivity detector. Prior to TPR measurements, 100 mg of the catalyst was activated by $H_2$ at 100 °C for 1 h and then cooled down to RT in Ar. Afterwards, the catalyst was subjected to 10 vol% $H_2$/Ar at a flow rate of 30 mL $min^{-1}$ and heated to 150 °C at 10 °C $min^{-1}$.

**Catalytic tests.** Selective hydrogenation of acetylene in excess ethylene was carried out in a fixed bed vertical quartz reactor, with a space velocity of 120,000 mL $h^{-1}$ $g^{-1}$. The reaction gas consisting of 1.0 vol% $C_2H_2$, 20.0 vol% $C_2H_4$, 20.0 vol.% $H_2$, and 59 vol.% $N_2$ was fed at a flow rate of 60 mL $min^{-1}$, simulating the front-end hydrogenation conditions. Typically, 30 mg of the catalyst (diluted by 400 mg of quartz sand) was activated by $H_2$ (20 mL $min^{-1}$) at 100 °C for 1 h and then cooled to RT in $N_2$ prior to the catalytic reaction. The gas components from the microreactor outlet were analyzed by online gas chromatography (GC; Shimadzu GC-2010) equipped with a flame ionization detector. It is important to note that the pretreatment in $H_2$ is important as the long-term exposure of $Pd_{1.0}$/$Bi_2O_3$/$TiO_2$ in air might oxidize Pd and demolish Pd–Bi bonding. Control experiment also suggests that untreated $Pd_{1.0}$/$Bi_2O_3$/$TiO_2$ has poor selectivity toward ethylene. To this end, all catalysts were pretreated in $H_2$ before catalytic measurements.

$C_2H_4$ and $C_2H_6$ were the only products detected by GC. Negligible oligomers were formed during the hydrogenation process, likely due to the short contact time and high concentration of hydrogen[10]. $C_2H_2$ conversion and $C_2H_4$ selectivity were calculated as:

$$C_2H_2 \text{ conversion} = \frac{[C_2H_2]_{inlet} - [C_2H_2]_{outlet}}{[C_2H_2]_{inlet}} \times 100\% \quad (1)$$

$$C_2H_4 \text{ selectivity} = \frac{[C_2H_2]_{inlet} - [C_2H_2]_{outlet} - [C_2H_6]_{outlet}}{[C_2H_2]_{inlet} - [C_2H_2]_{outlet}} \times 100\% \quad (2)$$

**Computational details.** In this work, Vienna Ab initio Simulation Package[63–66] was used to conduct density functional calculations within the generalized gradient approximation of RPBE functional developed by the Nørskov group[67]. Ionic cores and electrons were described by the projector augmented wave method[68,69]. The energy cutoff was set to be 500 eV and the force threshold was 0.05 eV $Å^{-1}$. We used constrained minimization method[59,70–72] to locate the transition state structures. The adsorption energies of $C_2H_x$ ($x$ = 2, 4, and 6) were calculated as: $E_{ads} = E_{C2Hx+slab} - (E_{C2Hx} + E_{slab})$, where $E_{C2Hx+slab}$ is the energy of the system after adsorption of $C_2H_x$ species, $E_{C2Hx}$ is the energy of the gas-phase $C_2H_x$ adsorbate, and $E_{slab}$ is the energy of the slab.

The optimized lattice parameters of $Bi_2O_3$ were $a$ = 5.981 Å, $b$ = 8.340 Å, and $c$ = 7.591 Å, which are close to experimental values (monoclinic, $P2_1/c(14)$, $a$ = 5.849 Å, $b$ = 8.166 Å, and $c$ = 7.510 Å). A $Bi_2O_3$(100) slab was built with a 2 × 2 supercell with nine atomic layers, and the Pd cluster structure was built by adding 8 Pd atoms onto the $Bi_2O_3$(100) surface, followed by structure optimization. To build a valid catalyst model of Pd cluster, we followed these rules: (i) Pd would disperse on the $Bi_2O_3$ surface with a size <2 nm; (ii) the structure should give rise to similar CNs listed in Supplementary Table 2; (iii) the model should be stable and would not deform under reaction condition, which has been widely discussed. During the optimizations involving Pd cluster structure, the bottom two layers of

atoms of $Bi_2O_3(100)$ component were fixed to simulate bulk structure of $Bi_2O_3$, while the other atoms were fully relaxed. The vacuum layer was set higher than 12 Å to avoid spurious interaction between adjacent slabs. The $k$-point grid used in the surface Brillouin zone was $1 \times 1 \times 1$ for all the calculations. More details about the structure development are provided in Supplementary Information (Supplementary Figs. 16 and 17).

## Data availability
All the data supporting the findings of this study are available within the article and its Supplementary Information files or from the corresponding authors upon reasonable request. Source data are provided with this paper.

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

## Acknowledgements

This work was financially supported by National Natural Science Foundation of China (92045301, 91545113, 91845203, 21802122, 21703050), China Postdoctoral Science Foundation (2019M662020), and Shell Global Solutions International B. V. (PT71423, PT74557). Y.Z. acknowledges financial support from the Zhejiang Provincial Natural Science Foundation of China (LR18B030003), National Natural Science Foundation of China (51701181), and the Thousand Talents Program for Distinguished Young Scholars. We thank the HPC Platform of ShanghaiTech University for computing time. This research used resources of the 8-BM and 7-BM Beamline of the National Synchrotron Light Source II, a U.S. Department of Energy (DOE) Office of Science User Facility operated for the DOE Office of Science by Brookhaven National Laboratory under Contract No. DE-SC0012704.

## Author contributions

J.F. and S.Z. designed the study. S.Z. and B.L. performed most of the experiments. W.Y., C.Z., Y.Z., and Y.W. performed the electron microscopy characterization. K.Y. and B.Y. finished the DFT calculations. L.M., Y.D., J.M., and Z.J. carried out the X-ray structure characterization and analysis. W.H. carried out the microcalorimetric measurements. Z.G. and Y.C. performed the in situ NAP-XPS studies. L.L., J.L., and L.X. performed the TPR and TPD measurements. S.Z., K.Y., J.F., B.Y., and Y.Z. wrote the paper. All authors interpreted the data and contributed to the preparation of the manuscript.

## Competing interests

The authors declare no competing interests.
