## [Peer Review File · Nature Communications]

Title: Grafting nanometer metal/oxide interface towards enhanced low-temperature acetylene semi-hydrogenationEditorial Note: This manuscript has been previously reviewed at another journal that is not operating a transparent peer review scheme. This document only contains reviewer comments and rebuttal letters for versions considered at *Nature Communications*.

REVIEWER COMMENTS

Reviewer #1 (Remarks to the Author):

In this paper, the author has synthesized a hybrid Pd/Bi₂O₃ cluster loaded on the TiO₂ support, which present good performance on the selective hydrogenation of acetylene to ethylene. Multiple characterizations have been applied to prove the hybrid structure, and the existence of intra-cluster electron transfer. Here I still have some question about this paper before publication.

1. The previous reviewers have referred my concern about the XPS and XAFS results, it seems the author had enough evidence to prove the interface structure of Pd and Bi₂O₃ now, while, I still doubt about the uniformity of this structure, as we can see from the elemental mapping of Pd_{1.0}Bi/TiO₂. there are some individual Pd clusters can be seen, which cannot be ignored by the author, this may also be the reason for the perfect performance of this catalyst for acetylene hydrogenation.

2. About the catalytic result, the author had compared the catalytic performances of different catalysts, Pd_{1.0}Bi/TiO₂, Pd_{0.2}Bi/TiO₂ and PdAg₃/Al₂O₃, where I have several doubts about this part. First, it's not advisable for the author to use a Al₂O₃ instead of TiO₂ loaded PdAg₃ catalyst for comparison. Second, based on Fig 3a, as far as I'm concerned, PdAg₃/Al₂O₃ catalyst was also a good catalyst for this reaction, it also shows excellent selectivity to C₂H₄, it's just the author raise the temperature to get a high C₂H₂ conversion, the selectivity of C₂H₄ at different C₂H₂ conversion was compared under different temperature, which makes it unfair to make comparison. We know that the hydrogenation reaction was related a lot with the temperature. Typically, the higher the temperature, the higher the hydrogenation rate. Third, there is no further characterization about the PdAg₃/Al₂O₃ catalyst. I doubt the electron structure of Pd was also controlled by Ag, It's just that compared to Bi, the hydrogenation ability of Pd is weakened more by Ag, which makes it less reactivity than Pd_{1.0}Bi/TiO₂ catalyst. From this point of view, the scientific novelty of this catalyst is weakened.

3. It seems the role of the interface is not reflected in the paper, except enrich the electron of Pd.

Meanwhile, based on the DFT, the reaction was happened on the surface of Pd, instead of the interface.

4. The author declaimed that the hybrid clusters feature an intra-cluster electron transfer towards Pd and enables much weaker ethylene bonding, more direct evidence should be given to prove this. For example, C₂H₄-pulse or C₂H₄ TPD (detected by MS). The microcalorimetric study cannot reflect both the adsorption strength and the adsorption capacity at the same time.

Reviewer #2 (Remarks to the Author):

Review Overview

In general, there is merit in the study and the potential for high impact, but the authors focus on a fine microscopic detail that is quite difficult to verify experimentally. Not that the focus is too fine, but that the phenomenon is hard to prove. Likewise, computationally proving the presence of a reduced metal-oxide interface is also not trivial given the role of material synthesis and atomic H spill-over effects.

Despite an extensive study, I remain unconvinced that the catalytic activity trends are dominantly caused by electron density transfer from Bi₂O₃ to Pd. There is a long history of scientists evoking a partially anionic metal produced by a reduced metal-oxide interface and considerable evidence in some subfields that reduced interfaces are not realistic under non-model catalyst synthesis and handling conditions and are dominantly produced and their catalytic activity observed in highly reducing conditions of UHV surface science studies. Nonetheless, given the reducibility of Bi₂O₃, the reducing conditions of semihydrogenation, and the experimental results, some steps have been made to motivate the production of partially anionic Pd via reduced Bi₂O₃. However, I have several reservations that are mostly focused on a lack of information provided by the authors, the possibility of a PdBi IMC exhibiting appropriate surface chemistry to perform the reaction, the possibility of Bi blocking sites on Pd, and the assumptions associated with producing the DFT model. On a positive note, the characterization and thorough understanding of catalyst science literature are admirable. However, there have been some omissions in the review of the role of Bi that must be addressed at least in short fashion, e.g., site blocking and PdBi IMC formation. In the end, without additional control studies and more realistic DFT models, it is hard to believe the fine details of the fundamental framework presented. The computational work may be easily improved. Experimental control runs also are critical. With these, the entire effort would be more convincing.

Specific Issues

- 1) The catalyst naming should be adjusted such that it is indicative of the materials composition, e.g., Pd/Bi₂O₃/TiO₂.
- 2) The EXAFS signal at 2.6Å also corresponds to Pd-Pd scattering. Are the authors ascribing this signal to Pd-Bi bonds as well? The oxidation test that illustrated the loss of this feature could simply be explained through the formation of PdO and the loss of Pd-Pd scattering paths. See: 10.1039/C8CP00517F.
- 3) The claim of a Pd-Bi bond could be motivated computationally by performing a surface termination study of Bi₂O₃. However, it is suggested that this analysis would show an oxygen-terminated Bi₂O₃ and that Pd is not in direct contact with Bi. However, this bond is likely not a critical feature in producing electron density polarization of Pd if the surface is heavily metal terminated. The extra electron density will go where it is most energetically favorable anyways.
- 4) The fluxionality of the small clusters even under XAS conditions may affect the coordination number. Performing these experiments under LN₂ temperatures would aid in limiting element mobility issues.
- 5) The catalyst naming convention is further convoluted once the authors start to discuss a PdBi IMC catalyst. The naming convention should be Pd/Bi₂O₃/TiO₂ for the main catalyst. The IMC catalyst is appropriately named PdBi/TiO₂.
- 6) The authors claim to purposefully produce PdBi IMC on TiO₂, yet no description of this synthesis is given. This must be described and is critical such that readers can determine if IMC formation is possible under either the synthesis of Pd/Bi₂O₃/TiO₂ or under reaction conditions.
- 7) Ultimately, the aim of the authors is to promote that they have Pd supported on a Bi-terminated

Bi₂O₃. Unfortunately, this claim is not indisputably supported by the characterization.

8) The authors discuss the shifts observed in the K-edge XANES in the context of d-state filling, yet L-edge is far more sensitive since promoted core electrons are populating d-states directly. The choice of the K-edge must be motivated.

9) The DFT model development has not been performed in an exhaustive fashion. It appears the model is motivated by the need to promote an idea rather than a clear connection with reality. Specifically, both top and bottom of the Bi₂O₃ model slab are Bi terminated producing a highly reduced model that will more readily allow for electron density transfer to Pd. The bottom of the slab should be stoichiometrically terminated by oxygen. The termination of the top of the slab needs to be determined via ab initio thermodynamics. If this were done, it is almost assured that the most promising termination is oxygen terminated. Then the authors could determine the reducibility of the surface using H₂ to motivate the production of an oxygen-lean surface termination. This more thorough computational study is critical since the experimental characterization does not clearly motivate an oxygen-lean Bi₂O₃ surface.

10) The literature support for highly reduced oxide surfaces promoting the reduction of supported metals needs to be scrutinized carefully. Many studies that promoted this view were performed under the naturally reducing conditions of UHV surface science. Many decades of follow-up studies aimed to understand the effect of non-model catalyst synthesis and pretreatment conditions and contact with the ambient. These studies motivated that oxygen-lean oxide surfaces are not stable and likely fully oxidized. Even a handful of UHV STM studies illustrated that oxygen-vacancy laden surfaces were not stable even under a well-baked UHV system condition and what were claimed to be oxygen vacancies were actually hydroxyl groups from H₂O dissociating rapidly on the reduced oxide surface. This study is unfortunately promoting a less than realistic metal-oxide interface and must work to thoroughly support their claim that the Bi₂O₃ surface is metal terminated. Unfortunately, it appears that they have not achieved this.

11) Calling the catalyst of focus a "model catalyst" is quite misleading. It is quite ill-defined and the source of its catalytic activity is still unclear.

12) The authors need to show the catalytic performance of the PdBi IMC catalyst for comparison.

13) The effect of Bi could be a simple site blocking mechanism. This seems to be an effect that is completely avoided in the discussion of the catalyst. Was this effect directly investigated? It could be by simply producing a Pd/TiO₂ catalyst and then adding Bi to the catalyst with an appropriate reduction pre-treatment procedure afterwards.

14) How is Bi oxidized to Bi₂O₃ in the synthesis method? There appears to be no elevated temperature oxidation or reduction pretreatment applied after initial catalyst synthesis.

15) No information is provided for sample handling before XPS, XRD, or XAS measurements. This information must be provided so the reader can predict the chemical state of the catalyst and how ambient may have affected composition.

16) At a minimum, the DFT model should be motivated significantly and the critical feature of surface termination be discussed directly. This is a vital feature of the model system and the electron density transfer to the metal. Maybe even add the oxygens to the bottom of the slab and show that the charge

transfer still occurs.

Reviewer #3 (Remarks to the Author):

This paper investigates a nanometer metal/oxide interface – hybrid Pd/Bi₂O₃ – in selective hydrogenation of acetylene, with promising catalytic performances.

An original structural model for the catalytic system is proposed, which is supported by several arguments, but which can also be discussed.

1- There are several possible phases for the 1:1 BiPd IMC, for which the shortest Pd-Bi distance ranges from 2.83 Å to 2.86 Å [Canadian Mineralogist 28 (1990) 751 ; J. Less Common Metals 64 (1979) P17 ; Soviet Physics JETP 5 (1957) 1064].

However, the authors talk about the Pd_{1.0}Bi/TiO₂-IMC, suggesting that they identified a specific phase, which is unfortunately not defined with the conventional way (space group and lattice parameters, instead of PDF#29-0238).

They invoked the shift in R-space (0.06 Å, Fig. S9b) to assess that the Pd-Bi distances are not the same in Pd_{1.0}Bi/TiO₂ and in Pd_{1.0}Bi/TiO₂-IMC. However, it does not exclude that two different IMCs phases are formed.

2- Information about the crystal structure of Bi₂O₃ is also lacking. There are several phases with the Bi₂O₃ stoichiometry (Mat. Letters 64 (2010) 2247-2250, Chem. Phys. Letters 378 (2003) 395-399, PRB 82 (2010) 024106, PRB 83 (2011) 214102, Acta Cryst. C44 (1988) 587-589) but the authors do not provide any information about the phase. Experimental lattice parameters are given in the section "Computational methods", but no reference is given, suggesting that it corresponds to the measurements done by the authors. However, this is not clear for me. In addition, the choice of the (100) orientation is not really well-argued. How is it consistent with the assertion that most Pd clusters are identified to bond and hybridize with Bi₂O₃ clusters without any fixed orientation relationship or facet preference?

3- According to the experimental measurements, the Bi-Pd distance is larger in Pd_{1.0}Bi/TiO₂ than in Pd_{1.0}Bi/TiO₂-IMC. If the assumptions made about the structures of Pd_{1.0}Bi/TiO₂ and Pd_{1.0}Bi/TiO₂-IMC are correct, different Bi-Pd distances calculated by DFT, using the model already built for Pd_{1.0}Bi/TiO₂ and a model that the authors can build for Pd_{1.0}Bi/TiO₂-IMC, based on the crystal structure determined by XRD, could support this assumption. Such a calculation would also provide information about the charge transfer in the Pd_{1.0}Bi/TiO₂ system and in the model built for Pd_{1.0}Bi/TiO₂-IMC, which can be useful to strengthen the analysis of the spectroscopic data.

4- Fig. 4b provides a reaction path for the hydrogenation reaction, (i) on the surface the Pd_{1.0}Bi/TiO₂

model provided by the authors and (ii) on Pd(111). The values of the barriers are given for Pd_{1.0}Bi/TiO₂ in the supplemental materials (Tab. S3), but not for Pd(111). The latter could have been compared with the literature, to assess the reliability of the computational method (constrained minimization method to locate the transition state structures).

General response: We sincerely thank the editor and all reviewers for their valuable feedback that we have based on to improve the quality of our manuscript. The reviewer comments are listed below in *italicized font* and specific concerns have been numbered. Our response is given point-by-point in normal font and changes/additions to the manuscript are given in **red text**. As suggested by the reviewer, we have changed the name of the catalysts. Bi/TiO₂, Pd_{1.0}Bi/TiO₂, Pd_{1.0}Bi/TiO₂-ox, and Pd_{1.0}Bi/TiO₂-IMC are renamed as Bi₂O₃/TiO₂, Pd_{1.0}/Bi₂O₃/TiO₂, Pd_{1.0}/Bi₂O₃/TiO₂-ox, and PdBi/TiO₂, respectively.

REVIEWER COMMENTS

Reviewer #1 (Remarks to the Author):

In this paper, the author has synthesized a hybrid Pd/Bi₂O₃ cluster loaded on the TiO₂ support, which present good performance on the selective hydrogenation of acetylene to ethylene. Multiple characterizations have been applied to prove the hybrid structure, and the existence of intra-cluster electron transfer. Here I still have some question about this paper before publication.

1. The previous reviewers have referred my concern about the XPS and XAFS results, it seems the author had enough evidence to prove the interface structure of Pd and Bi₂O₃ now, while, I still doubt about the uniformity of this structure, as we can see from the elemental mapping of Pd_{1.0}/Bi₂O₃/TiO₂. There are some individual Pd clusters can be seen, which cannot be ignored by the author, this may also be the reason for the perfect performance of this catalyst for acetylene hydrogenation.

Author response: We agree with the reviewer that there could be some individual Pd clusters in the Pd_{1.0}/Bi₂O₃/TiO₂. But the amount of these individual Pd clusters is quite small. The uniformity of the Pd-Bi₂O₃ hybrid clusters is evidenced by elemental mapping, ADF-STEM images, XAFS, and XPS.

(1) The concern of the existence of individual Pd clusters may due to the color selection for elemental mappings of Pd_{1.0}/Bi₂O₃/TiO₂ in **Fig. 1e**. Generally, Human vision is most sensitive to a green color. In the revised **Figs. 1e** (see below), we used green for Bi and red for Pd. In addition, more elemental mappings were presented in **Fig. S4**. As can be

seen from these figures, most Pd are spatially correlated to Bi, suggesting the Pd grafted Bi_2O_3 hybrid cluster structure. Individual Pd clusters in the $\text{Pd}_{1.0}/\text{Bi}_2\text{O}_3/\text{TiO}_2$ are quite rare.

(2) Careful inspection on several representative ADF-STEM images of $\text{Pd}_{1.0}/\text{Bi}_2\text{O}_3/\text{TiO}_2$ (**Figs. 1c, d and S5**) allows the discrimination of Pd- and Bi-containing hemi-clusters from their distinct contrasts for adjacent clusters. The usually smaller hemi-clusters featuring less bright contrast (marked by circle) are directly attached to larger and brighter ones assigned to Bi_2O_3 clusters, further confirming that Pd is grafted onto the surface of Bi_2O_3 clusters. Notably, most such Pd clusters are identified to bond and hybridize with Bi_2O_3 clusters without any fixed orientation relationship or facet preference, probably due to the ultra-small size and large strain of the clusters. Two representative ADF-STEM images of Pd- Bi_2O_3 nanoclusters with well-resolved Pd- and Bi-containing hemi-clusters are shown in **Figs. 1c and d**, of which the image contrasts and FFT patterns closely resemble the simulated ones of artificially constructed hybrid cluster models with different orientation relationship between two types of hemi-clusters made of Pd and $\alpha\text{-Bi}_2\text{O}_3$ structures respectively. The above observations unambiguously validate the proposed Pd- Bi_2O_3 hybrid structural model.

(3) Generally, due to the size effect, the XANES absorption edge of cluster samples will shift to higher energy (blue-shift) than that of foil (Ref.: *J. Phys. Chem. C* **2017**, 121, 361–374). In this study, however, $\text{Pd}_{1.0}/\text{Bi}_2\text{O}_3/\text{TiO}_2$ shows a marked red-shift of Pd K-edge XANES as compared with Pd foil and Pd/ TiO_2 (**Fig. 2e**). Similarly, $\text{Pd}_{1.0}/\text{Bi}_2\text{O}_3/\text{TiO}_2$ also exhibits a downshift of Pd 3d and 3p XPS that is opposite to the particle-size-induced shift (**Fig. S8**). The opposite trend arises from the Bi_2O_3 -to-Pd intra-cluster electron transfer and therefore suggests that most Pd clusters are grafted on the surface of Bi_2O_3 clusters.

Taking all these results together, it is conclusive that the individual Pd clusters are minority and only make minor contribution to the good performance of $\text{Pd}_{1.0}/\text{Bi}_2\text{O}_3/\text{TiO}_2$ for acetylene hydrogenation. As suggested by the reviewer, we discussed the presence and the influence of individual Pd clusters in the revised

manuscript as follows.

Followed by the secondary deposition of Pd, the clusters of Bi₂O₃ intergrown with Pd particles can be found according to the low-magnification ADF-STEM images (**Fig. S1**).

Careful inspection on several representative ADF-STEM images of Pd_{1.0}/Bi₂O₃/TiO₂ (**Figs. 1c, d and S5**) allows the discrimination of Pd- and Bi-containing hemi-clusters from their distinct contrasts for adjacent clusters. The usually smaller hemi-clusters featuring less bright contrast (marked by circle) are directly attached to larger and brighter ones assigned to Bi₂O₃ clusters, further confirming that Pd is grafted onto the surface of Bi₂O₃ clusters. Notably, most such Pd clusters are identified to bond and hybridize with Bi₂O₃ clusters without any fixed orientation relationship or facet preference, probably due to the ultra-small size and large strain of the clusters. Two representative ADF-STEM images of Pd-Bi₂O₃ nanoclusters with well-resolved Pd- and Bi-containing hemi-clusters are shown in **Figs. 1c and d**, of which the image contrasts and FFT patterns closely resemble the simulated ones of artificially constructed hybrid cluster models with different orientation relationship between two types of hemi-clusters made of Pd and α -Bi₂O₃ structures respectively. The above observations unambiguously validate the proposed Pd-Bi₂O₃ hybrid structural model. Quite rarely, such Pd clusters are observed to directly nucleate onto the TiO₂ substrate.

Fig. 1 | Microstructure of $\text{Pd}_{1.0}/\text{Bi}_2\text{O}_3/\text{TiO}_2$.

Fig. S4 | Elemental mapping of Pd_{1.0}/Bi₂O₃/TiO₂. a, b, c, d represent four different regions.

2. About the catalytic result, the author had compared the catalytic performances of different catalysts, Pd_{1.0}/Bi₂O₃/TiO₂, Pd_{0.2}/Bi₂O₃/TiO₂ and PdAg₃/Al₂O₃, where I have several doubts about this part. First, it's not advisable for the author to use a Al₂O₃ instead of TiO₂ loaded PdAg₃ catalyst for comparison. Second, based on Fig 3a, as far as I'm concerned, PdAg₃/Al₂O₃ catalyst was also a good catalyst for this reaction, it also shows excellent selectivity to C₂H₄, it's just the author raise the temperature to get a high C₂H₂ conversion, the selectivity of C₂H₄ at different C₂H₂ conversion was compared under different temperature, which makes it unfair to make comparison. We know that the hydrogenation reaction was related a lot with the temperature. Typically, the higher the temperature, the higher the hydrogenation rate. Third, there is no further characterization about the PdAg₃/Al₂O₃ catalyst. I doubt the electron structure of Pd was also controlled by Ag. It's just that compared to Bi, the hydrogenation ability of Pd is weakened more by Ag, which makes it less reactivity than Pd_{1.0}/Bi₂O₃/TiO₂ catalyst. From this point of view, the scientific novelty of this catalyst is weakened.

Author response: It is important to note that PdAg₃/Al₂O₃ is a benchmark catalyst for

acetylene hydrogenation. The composition of the PdAg₃/Al₂O₃ catalyst used in this study is the same as OleMax@251 (Ref.: U.S. patent 20200094226 A1), a widely used industrial catalyst for acetylene hydrogenation. Its structure and performance are well-illustrated in the literatures (Ref.: U.S. patent 20200094226 A1; *Chem. Commun.* **2013**, 49, 8350-8352; *J. Am. Chem. Soc.* 2021, DOI:10.1021/jacs.1c02471). That is why we did not present the structure characterization in the manuscript.

As suggested by the reviewer, we prepared PdAg₃/TiO₂ and tested its catalytic performance in acetylene hydrogenation. PdAg₃/TiO₂ shows similar catalytic performance as PdAg₃/Al₂O₃. Specifically, the C₂H₄ selectivity at 95% C₂H₂ conversion is ~ 10%, which is much lower than that of Pd_{1.0}/Bi₂O₃/TiO₂ (84.5%) and Pd_{0.2}/Bi₂O₃/TiO₂ (88.8%). It is important to highlight that the comparison with PdAg₃/Al₂O₃ is not meant to show the different promote effect of Bi and Ag. Instead, it helps us to evaluate the potential of Pd_{1.0}/Bi₂O₃/TiO₂ in industrial application. To compare the different promoting effect of Ag and Bi, we also synthesized Pd_{1.0}Ag/TiO₂ with the same method for the synthesis of Pd_{1.0}/Bi₂O₃/TiO₂. Catalytic test suggests that Pd_{1.0}Ag/TiO₂ over-hydrogenates acetylene to ethane at even room temperature, with a C₂H₄ selectivity of -167%. This result further proves the superior catalytic performance of Pd_{1.0}/Bi₂O₃/TiO₂ and their unique Pd-Bi₂O₃ hybrid-cluster structure.

The catalytic performance of PdAg₃/Al₂O₃ obtained in this study is similar to the literature results (Ref.: U.S. patent 20200094226 A1; *Chem. Commun.* **2013**, 49, 8350-8352). As noted by the reviewer, it is a good catalyst for this reaction, but the C₂H₄ selectivity at high C₂H₂ conversion still has room for improvement. In industry, the traces of acetylene in excess ethylene must be reduced to a very low level (typically < 15 ppm) before the downstream ethylene polymerization. A good hydrogenation catalyst should therefore be able to clean up C₂H₂ (typically, C₂H₂ conversion > 99%) with a good C₂H₄ selectivity. That's why we raise the reaction temperature of PdAg₃/Al₂O₃ and compare its C₂H₄ selectivity at a high C₂H₂ conversion. We agree with the reviewer that the hydrogenation reaction is related a lot with the temperature. **According to the convention of the catalysis studies (Ref.: *Science* **2008**, 320, 1320-1322; *ACS Catal.* **2016**, 6, 1054-1061; *Front. Chem. Sci. Eng.* **2015**, 9, 142-153), the**

C₂H₄ selectivity of different catalysts should be compared at the same C₂H₂ conversion rather than the same temperature. Typically, people plot C₂H₄ selectivity as a function of C₂H₂ conversion, as what we shown in **Fig. 3a**.

As suggested by the reviewer, we added the following text into the revised manuscript to illustrate why we choose PdAg₃/Al₂O₃ as a reference catalyst.

Both Pd/TiO₂ and well-established Pd₁Ag₃/Al₂O₃ catalysts^{28,43} were evaluated for comparison with Pd_{1.0}/Bi₂O₃/TiO₂. The composition and synthesis procedure of the PdAg₃/Al₂O₃ catalyst is the same as OleMax@251, a widely used industrial catalyst for acetylene hydrogenation⁴³.

3. It seems the role of the interface is not reflected in the paper, except enrich the electron of Pd. Meanwhile, based on the DFT, the reaction was happened on the surface of Pd, instead of the interface.

Author response: The interface plays important role in both the formation and the function of Pd-Bi₂O₃ hybrid nanoclusters. From a structural perspective, the unique interaction between interfacial Pd and Bi atoms allows the selective deposition of Pd onto Bi₂O₃ nanoclusters and therefore the formation of Pd-Bi₂O₃ hybrid clusters. While from a functional perspective, the Pd-Bi bonding and associated charge transfer at the interface allow the modulation of the electronic structure of Pd to optimize the bonding strength of reaction intermediates (e.g., ethylene). It is important to note that in this study, most Pd atoms are grafted on the surface of Bi₂O₃ clusters (please also refer to our reply to **Comment 1**). These interfacial Pd atoms are therefore essential parts of the interface. The hydrogenation reaction was happened on these interfacial Pd atoms and therefore reflected the role of the interface. The Pd₈-cluster model used in DFT calculations was constructed based on the experimental observations. It turns out that the interfacial Pd atoms are negatively charged and show weaker adsorption towards ethylene than Pd(111), which consequently leads to higher C₂H₄ selectivity.

In addition, we also synthesized Bi_{*x*}/Pd/TiO₂ (*x* = 0.5, 1, *x* is the molar ratio of Bi to Pd) by photo-depositing Bi onto 2.3 wt% Pd/TiO₂. Interestingly, both Bi_{0.5}/Pd/TiO₂ and Bi_{1.0}/Pd/TiO₂ exhibit 100% conversion of C₂H₂ and negative selectivity towards

ethylene (−123% for Bi_{0.5}/Pd/TiO₂ and −143% for Bi_{0.5}/Pd/TiO₂) at room temperature. These results suggest that a simple site blocking mechanism cannot explain the beneficial effect of Bi in this study. PdBi/TiO₂ composed of PdBi IMC (**Fig. S9**) shows 100% conversion of C₂H₂ and negative selectivity towards ethylene (−1319%) at room temperature. The strong exothermic effect of unselective acetylene hydrogenation eventually leads to a runaway temperature up to 58.5 °C. These results exclude PdBi IMC as the active site for Pd_{1.0}/Bi₂O₃/TiO₂ and further indicates the critical role of the nanometer Pd/Bi₂O₃ interface in the catalytic selectivity of Pd.

As suggested by the reviewer, we added more discussion to illustrate the importance of the interface.

4. The author declaimed that the hybrid clusters feature an intra-cluster electron transfer towards Pd and enables much weaker ethylene bonding, more direct evidence should be given to prove this. For example, C₂H₄-pulse or C₂H₄ TPD (detected by MS). The microcalorimetric study cannot reflect both the adsorption strength and the adsorption capacity at the same time.

Author response: As suggested by the reviewer, we performed C₂H₄-TPD (coupled with mass spectrometer) to investigate the adsorption mode of C₂H₄ over Pd/TiO₂ and Pd_{1.0}/Bi₂O₃/TiO₂. According to the literatures (Ref.: *J. Catal.* **2013**, 306, 146-154; *Catal. Today* **2012**, 185, 2-16; *Appl. Catal. A: Gen.* **2002**, 223, 161–172), the adsorption mode of ethylene can be identified by monitoring the mass signal of m/z = 27 in C₂H₄-TPD. Typically, there are three kinds of peaks in TPD patterns. The peak at 50 ~ 90 °C represents weakly adsorbed π-bonded ethylene, which consequently desorbs without decomposition. The peak centered at 105 ~ 150 °C represents di-σ-bonded ethylene, which undergoes decomposition followed by the recombination with hydrogen to produce ethylene as well as ethane. The peak above 250 °C corresponds to triply bound species, i.e., ethylidyne. As shown in **Fig. S12**, Pd/TiO₂ exhibits a large peak centered at ~ 115 °C, suggesting that it mainly adsorb ethylene via a strong di-σ-bonding. In the case of Pd_{1.0}/Bi₂O₃/TiO₂, the intensity of the peak at 115 °C is significantly reduced, accompanied by the emergence of a peak at ~ 65 °C. These results suggest that the

adsorption configuration of C_2H_4 is changed from strong di- σ -bonding to weak π -bonding, which likely originates from the low Pd-Pd coordination and the Bi_2O_3 -to-Pd intra-cluster electron transfer in $Pd_{1.0}/Bi_2O_3/TiO_2$. The change of adsorption mode eventually leads to reduced adsorption capacity of $Pd_{1.0}/Bi_2O_3/TiO_2$, as evidenced by the microcalorimetric study.

In addition to **Fig. S12**, the following discussion was added into the revised manuscript.

These observations can be attributed to the unique geometric and electronic structure of hybrid cluster, similar to the results reported in the alloying of Pd.^{47,48} The low Pd-Pd coordination and the Bi_2O_3 -to-Pd intra-cluster electron transfer likely change the adsorption configuration of C_2H_4 from stable ethylidyne to weak π -bonded C_2H_4 and promote the desorption of ethylene as the desired the product. To confirm this hypothesis, we further performed the temperature programmed desorption (TPD) of ethylene by monitoring the mass signal of $m/e = 27$ (**Fig. S12**). According to the literatures, the peak at ~ 65 °C could be assigned to weak π -bonded ethylene species, which readily desorb without decomposition.⁴⁹ The peak centered at ~ 115 °C originates from di- σ -bonded ethylene, which undergoes decomposition followed by the recombination of surface hydrocarbon species and hydrogen to produce ethylene and ethane.⁴⁹ Compared with Pd/TiO_2 , $Pd_{1.0}/Bi_2O_3/TiO_2$ exhibits a much weaker peak at ~ 115 °C but a significantly larger peak at 65 °C. These results confirm the adsorption configuration of C_2H_4 is changed from the strong σ -bonding for Pd/TiO_2 to weak π -bonding for $Pd_{1.0}/Bi_2O_3/TiO_2$.

Figure S12. C₂H₄-TPD over Pd/TiO₂ and Pd_{1.0}/Bi₂O₃/TiO₂.

Reviewer #2 (Remarks to the Author):

Review Overview

In general, there is merit in the study and the potential for high impact, but the authors focus on a fine microscopic detail that is quite difficult to verify experimentally. Not that the focus is too fine, but that the phenomenon is hard to prove. Likewise, computationally proving the presence of a reduced metal-oxide interface is also not trivial given the role of material synthesis and atomic H spill-over effects. Despite an extensive study, I remain unconvinced that the catalytic activity trends are dominantly caused by electron density transfer from Bi₂O₃ to Pd. There is a long history of scientists evoking a partially anionic metal produced by a reduced metal-oxide interface and considerable evidence in some subfields that reduced interfaces are not realistic under non-model catalyst synthesis and handling conditions and are dominantly produced and their catalytic activity observed in highly reducing conditions of UHV surface science studies. Nonetheless, given the reducibility of Bi₂O₃, the reducing conditions of semi-hydrogenation, and the experimental results, some steps have been made to motivate the production of partially anionic Pd via reduced Bi₂O₃. However, I have several reservations that are mostly focused on a lack of information provided by the authors, the possibility of a PdBi IMC exhibiting appropriate surface chemistry to

perform the reaction, the possibility of Bi blocking sites on Pd, and the assumptions associated with producing the DFT model. On a positive note, the characterization and thorough understanding of catalyst science literature are admirable. However, there have been some omissions in the review of the role of Bi that must be addressed at least in short fashion, e.g., site blocking and PdBi IMC formation. In the end, without additional control studies and more realistic DFT models, it is hard to believe the fine details of the fundamental framework presented. The computational work may be easily improved. Experimental control runs also are critical. With these, the entire effort would be more convincing.

General response: We sincerely thank the reviewer for valuable comments and suggestions. In the revised manuscript, we have made considerable effort to address the proposed issues from both experimental and theoretical perspectives. The point-by-point response to the reviewer's comments are listed below.

Specific Issues

1) The catalyst naming should be adjusted such that it is indicative of the materials composition, e.g., Pd/Bi₂O₃/TiO₂.

Author response: As suggested by the reviewer, we have changed the name of Pd_{1.0}Bi/TiO₂ to Pd_{1.0}/Bi₂O₃/TiO₂ in the revised manuscript and supporting information.

2) The EXAFS signal at 2.6 Ang also corresponds to Pd-Pd scattering. Are the authors ascribing this signal to Pd-Bi bonds as well? The oxidation test that illustrated the loss of this feature could simply be explained through the formation of PdO and the loss of Pd-Pd scattering paths. See: 10.1039/C8CP00517F.

Author response: It is important to highlight that EXAFS is an element-specific analysis because the energy required to excite a core electron is determinate of the element being analyzed. In this manuscript, the discussion of the signal at 2.6 Å is focused on the Bi L3-edge EXAFS spectra of Pd_{1.0}/Bi₂O₃/TiO₂. It reflects only the coordination environment of Bi rather than Pd. Therefore, this signal does not correspond to Pd-Pd scattering. The loss of this feature in the oxidation test cannot be

explained through formation of PdO and the loss of Pd-Pd scattering.

3) The claim of a Pd-Bi bond could be motivated computationally by performing a surface termination study of Bi₂O₃. However, it is suggested that this analysis would show an oxygen-terminated Bi₂O₃ and that Pd is not in direct contact with Bi. However, this bond is likely not a critical feature in producing electron density polarization of Pd if the surface is heavily metal terminated. The extra electron density will go where it is most energetically favorable anyways.

Author response: We sincerely thank the reviewer for valuable comments and suggestions. In the revised manuscript, we added more experimental evidence and DFT calculations to illustrate the rationality and the origin of the Pd-Bi bond.

(1) Experimental. It is important to note that the Pd-Bi₂O₃ hybrid clusters model was constructed mainly based on experimental observations. The Pd-Bi bond is evidenced by the characteristic peak at ~ 2.6 Å of Bi L3 EXAFS. The Pd-Bi bond length obtained from the fitting of Pd K and Bi L3 edge is 2.79 ± 0.03 Å, which is significantly smaller than the Pd-Bi bond length in Pd-O-Bi moieties (~ 3.5 Å). This result suggests that the observed spatial correlation of Pd-Bi pairs arises from the direct bonding between Pd- and Bi-terminated clusters. We agree with the reviewer that oxygen-terminated Bi₂O₃ is generally more stable than Bi-terminated Bi₂O₃. Pd directly deposited onto Bi₂O₃ likely bonds with O rather than Bi. To confirm this hypothesis, we synthesized a Pd/Bi₂O₃ sample by directly depositing Pd onto commercial Bi₂O₃ support. Interestingly, no characteristic peak at ~ 2.6 Å was observed in the Bi L3 EXAFS of Pd/Bi₂O₃, excluding the presence of Pd-Bi bond (please see below the new **Figure 2c**). In this study, Pd is in direct contact with Bi mainly because of the unique synthetic procedure of Pd_{1.0}/Bi₂O₃/TiO₂. Pd_{1.0}/Bi₂O₃/TiO₂ was synthesized by a stepwise photodeposition method, followed by a mild oxidation of Bi in the air. In brief, Bi³⁺ was reduced by the photogenerated electrons to produce Bi⁰ clusters on TiO₂. Subsequently, Pd was deposited onto Bi/TiO₂ to form Pd/Bi/TiO₂. When the sample was exposed to the air, Bi was spontaneously oxidized into Bi₂O₃ because the Gibbs free energy of the reaction

$(2\text{Bi(s)} + 1.5\text{O}_2\text{(g)} = \text{Bi}_2\text{O}_3\text{(s)} (298\text{--}544\text{ K}), \Delta G^\ominus (\text{J mol}^{-1}) = -574887 + 99.71T)$ is minus. The procedure was schematically illustrated in the new **Figure 1a**. The Pd-Bi bond preserves during the mild oxidation of Bi to Bi_2O_3 at room temperature occurred after the deposition of Pd onto Bi/TiO_2 , as evidenced by the characteristic Bi-Pd peak in Bi L3 EXAFS (**Fig. 2c**). More interestingly, 36% of this peak preserves even after oxidation in air at a higher temperature of $150\text{ }^\circ\text{C}$ ($\text{Pd}_{1.0}/\text{Bi}_2\text{O}_3/\text{TiO}_2\text{-ox}$). These results clearly indicate the good stability of Pd-Bi bond in $\text{Pd}_{1.0}/\text{Bi}_2\text{O}_3/\text{TiO}_2$ and suggest Pd supported on Bi terminated Bi_2O_3 as the structural model for $\text{Pd}_{1.0}/\text{Bi}_2\text{O}_3/\text{TiO}_2$ in the hydrogenation reaction conditions.

Figure 1a. Schematic illustration of the synthetic procedures.

Figure 2c. Fourier transform spectra of Bi L3-edge EXAFS. $\text{Pd}/\text{Bi}_2\text{O}_3$ was synthesized

by depositing 2 wt.% Pd onto Bi₂O₃. Please see the experimental details in supporting information.

(2) **DFT calculations.** As suggested by the reviewer, we calculated surface energies of O- and Bi- terminated Bi₂O₃(100) surfaces, which are listed in **Table R1**. It turns out that O-termination is thermodynamically more stable than Bi-termination. This is consistent with the argument of this reviewer and the experiments further performed in our work.

Table R1. Surface energy of low-index surfaces and terminations on Bi₂O₃.

Surfaces and terminations	Surface energy (eV/Å ²)
Bi-terminated (100)	0.0978
O-terminated (100)	0.0587

As suggested by the reviewer in **Comment 9**, we further calculated the free energy change of removing surface oxygen atoms on O-terminated (100) surfaces with hydrogen to form a Bi-terminated surface. In the calculations, we utilized a 11-layer 1×1 Bi₂O₃(100) surface consisting of 44 Bi and 66 O atoms to represent an O-exposed surface. Then we removed the 6 surface O atoms to obtain the structure exposing surface Bi atoms. The free energy change (ΔG) can be calculated as follows:

$$\Delta G = (G_{\text{Bi}_{44}\text{O}_{60}} + 6(G_{\text{H}_2\text{O}} - G_{\text{H}_2}) - G_{\text{Bi}_{44}\text{O}_{66}})/N$$

where the G of solid materials is equal to their corresponding DFT-energy (E). The free energies of H₂O and H₂ were calculated at 40 °C, which is close to the synthesis temperature of Pd_{1.0}/Bi₂O₃/TiO₂ and acetylene hydrogenation temperature. The entropy of gaseous species was obtained from NIST database (<https://cccbdb.nist.gov/introx.asp>). N is the number of O atoms that have been removed during the process and it is 6. It was found that the average ΔG is negative, i.e., -1.81 eV when removing the surface O atom, showing that the O atoms on the (100) surface are readily removed under reaction conditions, and the Bi-layer would be exposed. Therefore, we used the Bi-terminated Bi₂O₃(100) for further studies.

In terms of the electron transfer between Bi and Pd, we also added more DFT

calculations (please also refer to **Comment 9** for details). Following the reviewer's suggestion in **Comment 9**, we considered a Bi_2O_3 slab where its bottom is terminated by O and the top terminated by Bi. The new Bi_2O_3 - Pd_8 structure is shown in **Fig. S16**. In addition, we used the model in **Fig. S16** to conduct Bader charge analysis, and the data are given in **Fig. S17**. One can find from the figure that the charge-transfer effect of the new Bi_2O_3 - Pd_8 cluster structure follows the same trend as we proposed in the original manuscript (see **Fig. S11**), i.e., the Bi_2O_3 transfers electrons to Pd.

Fig. S16. Structure of the new Pd_8 -model with its bottom surface terminated by O. The number of Bi and O atoms follow the stoichiometric ratio of Bi_2O_3 .

No. of Pd atom	Charge /e
1	-0.26
2	-0.26
3	-0.01
4	-0.32
5	-0.12
6	-0.24
7	-0.15
8	-0.32
-0.21 (Average)	

Fig. S17. Bader charge of Pd atoms on the Pd_8 -cluster model with O-terminated bottom slab. A negative charge here means the atom withdraws electrons from other atoms.

Part of the above discussion was added into the supporting information to illustrate the

model development. In addition to new **Figs. 1a, 2c, S16 and S17**, we also added the following discussion into the revised manuscript.

In brief, Bi^{3+} was reduced by the photogenerated electrons to produce Bi^0 clusters on TiO_2 . Subsequently, Pd was deposited onto Bi/TiO_2 to form $\text{Pd}/\text{Bi}/\text{TiO}_2$. When the sample was exposed to the air, Bi was spontaneously oxidized into Bi_2O_3 because the Gibbs free energy of the reaction is minus. The procedure was schematically illustrated in the **Fig. 1a**.

Photo-deposition procedure is critical to ensure the formation of direct Pd-Bi bonding in $\text{Pd}/\text{Bi}_2\text{O}_3$ clusters as depicted in **Fig. 1a**. Theoretically, Bi_2O_3 favours an oxygen-termination. Pd supported on Bi_2O_3 would be in direct contact with O rather than Bi. We prepared a $\text{Pd}/\text{Bi}_2\text{O}_3$ sample by directly depositing Pd onto commercial Bi_2O_3 support. Interestingly, no characteristic peak at $\sim 2.6 \text{ \AA}$ was observed in the Bi L3 EXAFS of $\text{Pd}/\text{Bi}_2\text{O}_3$, excluding the presence of Pd-Bi bond (**Fig. 2c**). In this study, photo-deposition procedure ensures the Pd cluster deposited on reduced Bi^0 clusters, and allows the formation of Pd-Bi bonding during the synthesis as depicted in Figure 1. The Pd-Bi bond preserves during the mild oxidation of Bi to Bi_2O_3 at room temperature occurred after the deposition of Pd onto Bi/TiO_2 , as evidenced by the characteristic Bi-Pd peak in Bi L3 EXAFS (**Fig. 2c**). More interestingly, 36% of this peak preserves even after oxidation in air at a higher temperature of $150 \text{ }^\circ\text{C}$ ($\text{Pd}_{1.0}/\text{Bi}_2\text{O}_3/\text{TiO}_2\text{-ox}$). These results clearly indicate the good stability of Pd-Bi bond in $\text{Pd}_{1.0}/\text{Bi}_2\text{O}_3/\text{TiO}_2$ and suggest Pd supported on Bi terminated Bi_2O_3 as the structural model for $\text{Pd}_{1.0}/\text{Bi}_2\text{O}_3/\text{TiO}_2$ in the hydrogenation reaction conditions.

Model of Bi_2O_3 supported Pd_8 cluster catalyst was built according to the experimental characterization results, and the structure of which is shown in **Fig. 4a** (please see the details of model development in supporting information).

4) The fluxionality of the small clusters even under XAS conditions may affect the coordination number. Performing these experiments under LN_2 temperatures would aid in limiting element mobility issues.

Author response: We sincerely thank the reviewer for raising this technique issue. We

agree with the reviewer that performing XAS experiments under LN₂ temperatures is beneficial. But it won't directly affect the coordination number. Instead, it reduces the contribution of the Debye-Waller factor, which contains information on thermal disorder and static disorder (*Ref.*: Geantet, C. and Pichon, C. (2012). X-Ray Absorption Spectroscopy. In *Characterization of Solid Materials and Heterogeneous Catalysts* (eds M. Che and J.C. Védrine)). Owing to the interaction between clusters and support, the supported clusters have a relatively stable and less fluxional structure. The XAS experiments were therefore performed at room temperature, as frequently reported in the literatures (*J. Am. Chem. Soc.* 2018, 140, 41, 13142–13146; *Adv. Mater.* 2019, 31, 1900509; *Angew. Chem. Int. Ed.*, 2015, 54, 11265-11269).

5) *The catalyst naming convention is further convoluted once the authors start to discuss a PdBi IMC catalyst. The naming convention should be Pd/Bi₂O₃/TiO₂ for the main catalyst. The IMC catalyst is appropriately named PdBi/TiO₂.*

Author response: As suggested by the reviewer, we have changed the name of the catalysts. The main catalyst is named Pd_{1.0}/Bi₂O₃/TiO₂ and the IMC catalyst is named PdBi/TiO₂ in the revised manuscript.

6) *The authors claim to purposefully produce PdBi IMC on TiO₂, yet no description of this synthesis is given. This must be described and is critical such that readers can determine if IMC formation is possible under either the synthesis of Pd/Bi₂O₃/TiO₂ or under reaction conditions.*

Author response: As suggested by the reviewer, we added the details of the preparation of PdBi/TiO₂-IMC in the supporting information as follows.

Synthesis of PdBi/TiO₂-IMC. PdBi/TiO₂ was prepared by a two-step deposition-reduction method using NaBH₄ as the reducing agent. Typically, 100 mg of TiO₂ and 11.6 mg of Bi(NO₃)₃·5H₂O were dispersed in 8 mL of ethylene glycol aqueous solution (50 vol.%). The suspension was stirred for 90 min before the dropwise addition of 2 mL of NaBH₄ aqueous solution (5 M). After stirred for another 30 min, 4 mL of acetone dissolving 5.4 mg of Pd(OAc)₂ was added to above suspension dropwise. The mixture

was stirred for 1 h to allow a complete reduction of Pd. The precipitates were collected by centrifugation, washed twice by water and ethanol, and then dried in a vacuum oven at 40 °C. The catalysts were treated in H₂/Ar at 350 °C for 2 h before used for characterizations and catalytic tests.

7) Ultimately, the aim of the authors is to promote that they have Pd supported on a Bi-terminated Bi₂O₃. Unfortunately, this claim is not indisputably supported by the characterization.

Author response: We sincerely thank the reviewer for raising this important question. As extensively discussed in **Comment 3**, the direct Pd-Bi bond is evidenced by the characteristic peak at ~ 2.6 Å of Bi L3 EXAFS. The Pd-Bi bond length obtained from the fitting of Pd K and Bi L3 edge is 2.79 ± 0.03 Å, which is significantly smaller than the Pd-Bi bond length in Pd-O-Bi moieties (~ 3.5 Å). This result suggests that the observed spatial correlation of Pd-Bi pairs arises from the direct bonding between Pd- and Bi-terminated clusters. To confirm this result, we synthesized a Pd/Bi₂O₃ sample for comparison. The Pd/Bi₂O₃ was prepared by directly depositing Pd onto oxygen-terminated Bi₂O₃ support. Interestingly, no characteristic peak at ~ 2.6 Å was observed in the Bi L3 EXAFS of Pd/Bi₂O₃ (**Fig. 2c**), which further confirms that the peak at ~ 2.6 Å is a characteristic peak for Pd-Bi bond. In this study, Pd is in direct contact with Bi mainly because of the unique synthetic procedure of Pd_{1.0}/Bi₂O₃/TiO₂. Pd_{1.0}/Bi₂O₃/TiO₂ was synthesized by a stepwise photo-deposition method, followed by a mild oxidation of Bi in the air. In brief, Bi³⁺ was reduced by the photogenerated electrons to produce Bi⁰ clusters on TiO₂. Subsequently, Pd was deposited onto Bi/TiO₂ to form Pd/Bi/TiO₂. When the sample was exposed to the air, Bi was spontaneously oxidized into Bi₂O₃ because the Gibbs free energy of the reaction ($2\text{Bi}(\text{s}) + 1.5\text{O}_2(\text{g}) = \text{Bi}_2\text{O}_3(\text{s})$ (298–544 K), ΔG^\ominus (J mol⁻¹) = $-574887 + 99.71T$) is minus. The procedure was schematically illustrated in the new **Fig. 1a**. The Pd-Bi bond preserves during the mild oxidation of Bi to Bi₂O₃ at room temperature occurred after the deposition of Pd onto Bi/TiO₂, as evidenced by the characteristic Bi-Pd peak in Bi L3 EXAFS (**Fig. 2c**). More interestingly, 36% of this peak preserves even after oxidation in air at a higher

temperature of 150 °C ($\text{Pd}_{1.0}/\text{Bi}_2\text{O}_3/\text{TiO}_2\text{-ox}$). These results clearly indicate the good stability of Pd-Bi bond in $\text{Pd}_{1.0}/\text{Bi}_2\text{O}_3/\text{TiO}_2$ and suggest Pd supported on a Bi terminated Bi_2O_3 as the structural model for $\text{Pd}_{1.0}/\text{Bi}_2\text{O}_3/\text{TiO}_2$ in the hydrogenation reaction conditions. Taking all these results together, the structure models used in DFT calculations are reliable.

As suggested by the reviewer, the following discussion was added into the revised manuscript.

In brief, Bi^{3+} was reduced by the photogenerated electrons to produce Bi^0 clusters on TiO_2 . Subsequently, Pd was deposited onto Bi/TiO_2 to form $\text{Pd}/\text{Bi}/\text{TiO}_2$. When the sample was exposed to the air, Bi was spontaneously oxidized into Bi_2O_3 because the Gibbs free energy of the reaction is minus. The procedure was schematically illustrated in the **Fig. 1a**.

Photo-deposition procedure is critical to ensure the formation of direct Pd-Bi bonding in $\text{Pd}/\text{Bi}_2\text{O}_3$ clusters as depicted in **Fig. 1a**. Theoretically, Bi_2O_3 favours an oxygen-termination. Pd supported on Bi_2O_3 would be in direct contact with O rather than Bi. We prepared a $\text{Pd}/\text{Bi}_2\text{O}_3$ sample by directly depositing Pd onto commercial Bi_2O_3 support. Interestingly, no characteristic peak at $\sim 2.6 \text{ \AA}$ was observed in the Bi L3 EXAFS of $\text{Pd}/\text{Bi}_2\text{O}_3$, excluding the presence of Pd-Bi bond (**Fig. 2c**). In this study, photo-deposition procedure ensures the Pd cluster deposited on reduced Bi^0 clusters, and allows the formation of Pd-Bi bonding during the synthesis as depicted in Figure 1. The Pd-Bi bond preserves during the mild oxidation of Bi to Bi_2O_3 at room temperature occurred after the deposition of Pd onto Bi/TiO_2 , as evidenced by the characteristic Bi-Pd peak in Bi L3 EXAFS (**Fig. 2c**). More interestingly, 36% of this peak preserves even after oxidation in air at a higher temperature of 150 °C ($\text{Pd}_{1.0}/\text{Bi}_2\text{O}_3/\text{TiO}_2\text{-ox}$). These results clearly indicate the good stability of Pd-Bi bond in $\text{Pd}_{1.0}/\text{Bi}_2\text{O}_3/\text{TiO}_2$ and suggest Pd supported on Bi terminated Bi_2O_3 as the structural model for $\text{Pd}_{1.0}/\text{Bi}_2\text{O}_3/\text{TiO}_2$ in the hydrogenation reaction conditions.

Model of Bi_2O_3 supported Pd_8 cluster catalyst was built according to the experimental characterization results, and the structure of which is shown in **Fig. 4a** (please see the details of model development in supporting information).

8) The authors discuss the shifts observed in the K-edge XANES in the context of d-state filling, yet L-edge is far more sensitive since promoted core electrons are populating d-states directly. The choice of the K-edge must be motivated.

Author response: We agree with the reviewer that the K-edge XANES should not be discussed in the context of d-state filling. In the revised manuscript, the discussion of Pd K-edge XANES was revised as follows.

Specifically, the Pd K-edge XANES profile of Pd/TiO₂ is very similar to that of Pd foil. In contrast, Pd_{1.0}/Bi₂O₃/TiO₂ shows a marked red-shift of the absorption edge energy and decrease in the white line intensity. This indicates the electron-richness of Pd atoms in Pd_{1.0}/Bi₂O₃/TiO₂ compared to those in Pd foil and Pd/TiO₂, which most likely arises from the Bi₂O₃-to-Pd intra-cluster electron transfer.

Besides, as suggested by the reviewer, we also collected Pd L3-edge XANES data at 8-BM of NSLS-II. According to the literatures (Ref.: *Catalysis Letters* 53 (1998) 155–159; *J. Phys. Chem. C* 2009, 113, 34, 15140–15147), the Pd L3 near-edge absorption peak at ~3173 eV indicates the excitation of 2p_{3/2} electrons to the vacant 4d-states. A stronger absorption peak is expected for catalysts containing Pd atoms with more unfilled 4d-states such that excitations of electrons to these states are highly probable. As can be seen from **Fig. S10**, Pd_{1.0}/Bi₂O₃/TiO₂ shows a much weaker Pd L3 white line intensity at ~3173 eV than Pd/TiO₂, indicating an enhanced d-band filling. Such phenomenon is similarly predicted by the Bader charge analysis over Pd₈ clusters supported by the Bi₂O₃ cluster as shown in **Fig. S11**. Interestingly, once Pd_{1.0}/Bi₂O₃/TiO₂ was mildly oxidized in air at 150 °C to Pd_{1.0}/Bi₂O₃/TiO₂-ox, the Pd L3 XANES peak becomes similar to that of Pd/TiO₂. This result confirms that the mild oxidation of Pd_{1.0}/Bi₂O₃/TiO₂ changes the Pd-Bi interface.

Fig. S10 Pd L3-edge XANES spectra for Pd/TiO₂, PdBi/TiO₂-IMC, Pd_{1.0}/Bi₂O₃/TiO₂ and Pd_{1.0}/Bi₂O₃/TiO₂-ox. Pd L3 near-edge absorption peak at ~3173 eV indicates the excitation of 2p_{3/2} electrons to the vacant 4d-states. A stronger absorption peak is expected for catalysts containing Pd atoms with more unfilled 4d-states.

In addition to new **Fig. S10**, we also included the following discussion into the manuscript.

Moreover, Pd_{1.0}/Bi₂O₃/TiO₂ shows a much weaker Pd L3 white line intensity at ~3173 eV than Pd/TiO₂ (**Fig. S10**), which indicates an enhanced d-band filling.

9) *The DFT model development has not been performed in an exhaustive fashion. It appears the model is motivated by the need to promote an idea rather than a clear connection with reality. Specifically, both top and bottom of the Bi₂O₃ model slab are Bi terminated producing a highly reduced model that will more readily allow for electron density transfer to Pd. The bottom of the slab should be stoichiometrically terminated by oxygen. The termination of the top of the slab needs to be determined via ab initio thermodynamics. If this were done, it is almost assured that the most promising termination is oxygen terminated. Then the authors could determine the reducibility of the surface using H₂ to motivate the production of an oxygen-lean surface termination. This more thorough computational study is critical since the experimental*

characterization do not clearly motivate an oxygen-lean Bi₂O₃ surface.

Author response: We sincerely thank the reviewer for valuable comments and suggestions. We developed the Pd₈-cluster model based on the idea that the model used in calculations should be consistent with experimental observations. The obtained interatomic distance between Pd and Bi on Pd₈-cluster structure is found to be similar with experimental values. Hence the model successfully represents the local Pd-Bi environment in the experiment. Furthermore, the reason we used the slab without O-occupied bottom terminations was to keep the surface layer identical to the bottom layer and to avoid possible effect of dipole moment on energy calculations.

As suggested by the reviewer, we also calculated the Bi₂O₃ structure with the O-terminated bottom slab (please also refer to **Comment 3**). The structure is given in **Fig. S16**, and the electron-transfer result is given in **Fig. S17**. One can find from the figure that the charge-transfer effect of the new Bi₂O₃-Pd₈ cluster structure follows the same trend as we proposed in the original manuscript (see **Fig. S11**), i.e., the Bi₂O₃ transfers electrons to Pd. In addition, we further compared the stability between the stoichiometric Bi₂O₃ and Oxygen-lean Bi₂O₃ under reducing conditions. The transition between the stoichiometric Bi₂O₃ (Bi_xO_{1.5x}) to oxygen-lean Bi₂O₃ (Bi_xO_y) structure can be regarded as the reaction:

And the free energy free of the reaction can be obtained by:

$$\Delta G = (1.5x - y)G_{\text{H}_2\text{O}} + G_{\text{Bi}_x\text{O}_y} - (1.5x - y)G_{\text{H}_2} - G_{\text{Bi}_x\text{O}_{1.5x}}$$

Using the above methods, we found it is spontaneous to reduce the O-terminated layer of stoichiometric Bi₂O₃(100) surface with H₂. The result demonstrates the stability of the Bi-terminated Bi₂O₃(100) surface under reducing conditions.

As suggested by the reviewer, we added more DFT calculations and discussion into the revised supporting information.

10) The literature support for highly reduced oxide surfaces promoting the reduction of supported metals needs to be scrutinized carefully. Many studies that promoted this view were performed under the naturally reducing conditions of UHV surface science. Many decades of follow-up studies aimed to understand the effect of non-model catalyst synthesis and pretreatment conditions and contact with the ambient. These studies

motivated that oxygen-lean oxide surfaces are not stable and likely fully oxidized. Even a handful of UHV STM studies illustrated that oxygen-vacancy laden surfaces were not stable even under a well-baked UHV system condition and what were claimed to be oxygen vacancies were actually hydroxyl groups from H₂O dissociating rapidly on the reduced oxide surface. This study is unfortunately promoting a less than realistic metal-oxide interface and must work to thoroughly support their claim that the Bi₂O₃ surface is metal terminated. Unfortunately, it appears that they have not achieved this.

Author response: We sincerely thank the reviewer for valuable comments. It helps us to understand the origin of Pd-Bi bonding in Pd_{1.0}/Bi₂O₃/TiO₂. As suggested by the reviewer, we added more experimental evidence and DFT calculations to support the direct Pd-Bi bonding in Pd_{1.0}/Bi₂O₃/TiO₂. Please also refer to **Comments 3, 7, and 9**. In this study, the direct Pd-Bi bonding likely results from the unique synthesis procedures. Because the mild oxidation of Bi to Bi₂O₃ occurred after the deposition of Pd onto Bi/TiO₂, the Pd-Bi bond preserves.

11) Calling the catalyst of focus a "model catalyst" is quite misleading. It is quite ill-defined and the source of its catalytic activity is still unclear.

Author response: As suggested by the reviewer, we have changed the name of the catalysts in the revised manuscript. The main catalyst is named Pd_{1.0}/Bi₂O₃/TiO₂ to illustrate that Pd is in contact with Bi₂O₃ to form hybrid clusters. The source of its catalytic activity is now clearly related to the Pd-Bi₂O₃ hybrid clusters.

12) The authors need to show the catalytic performance of the PdBi IMC catalyst for comparison.

Author response: As suggested by the reviewer, we showed the catalytic performance of the PdBi IMC catalyst in the revised supporting information and mentioned it in the revised manuscript as follows.

PdBi/TiO₂ composed of PdBi IMC (Fig. S9) shows 100% conversion of C₂H₂ and negative selectivity towards ethylene (-1319%) at room temperature. The strong exothermic effect of unselective acetylene hydrogenation eventually leads to a runaway

temperature up to 58.5 °C. These results exclude PdBi IMC as the active site for Pd_{1.0}/Bi₂O₃/TiO₂ and further indicates the critical role of the nanometer Pd/Bi₂O₃ interface in the catalytic selectivity of Pd.

13) *The effect of Bi could be a simple site blocking mechanism. This seems to be an effect that is completely avoided in the discussion of the catalyst. Was this effect directly investigated? It could be by simply producing a Pd/TiO₂ catalyst and then adding Bi to the catalyst with an appropriate reduction pre-treatment procedure afterwards.*

Author response: It is important to note that in the synthesis of Pd_{1.0}/Bi₂O₃/TiO₂, Bi was deposited onto TiO₂ before the deposition of Pd. The reduction pretreatment temperature is also too low to (only 100 °C) drive the migration of Bi₂O₃ onto Pd. Therefore, it is very unlikely that Bi will block the surface of Pd. To directly exclude the site blocking mechanism, we synthesized reference samples according to the suggestion of reviewer. The reference samples were synthesized by photodepositing Bi onto 2.3 wt% Pd/TiO₂ and pretreated at the same condition of Pd_{1.0}/Bi₂O₃/TiO₂ (reduction in H₂/Ar at 100 °C for 1 h). The samples were denoted as Bi_{*x*}/Pd/TiO₂ (*x* = 0.5, 1), where *x* is the Bi to Pd molar ratio. Both Bi_{0.5}/Pd/TiO₂ and Bi_{1.0}/Pd/TiO₂ exhibit 100% conversion of C₂H₂ and negative selectivity towards ethylene (-123% for Bi_{0.5}/Pd/TiO₂ and -143% for Bi_{1.0}/Pd/TiO₂) at room temperature. The strong exothermic effect of unselective acetylene hydrogenation eventually leads to a runaway temperature up to ~ 31 °C. These results suggest that a simple site blocking mechanism cannot explain the beneficial effect of Bi in this study. We sincerely thank the reviewer for valuable suggestion. Blocking Pd sites by surface Bi₂O₃ coverage or isolating Pd in PdBi intermetallic might be a good way to improve the catalytic performance of Pd catalysts. This will be our next project.

As suggested by the reviewer, the following discussion was added into the revised manuscript.

In addition, we also synthesized Bi_{*x*}/Pd/TiO₂ (*x* = 0.5, 1, *x* is the molar ratio of Bi to Pd) by photodepositing Bi onto 2.3 wt% Pd/TiO₂. Interestingly, both Bi_{0.5}/Pd/TiO₂ and Bi_{1.0}/Pd/TiO₂ exhibit 100% conversion of C₂H₂ and negative selectivity towards

ethylene (−123% for Bi_{0.5}/Pd/TiO₂ and −143% for Bi_{0.5}/Pd/TiO₂) at room temperature. These results suggest that a simple site blocking mechanism cannot explain the beneficial effect of Bi in this study.

14) How is Bi oxidized to Bi₂O₃ in the synthesis method? There appears to be no elevated temperature oxidation or reduction pretreatment applied after initial catalyst synthesis.

Author response: In this study, Bi was mildly oxidized to Bi₂O₃ when the sample was exposed to the air. As suggested by the reviewer, we revised Fig. 1a and added more discussion to illustrate the oxidation of Bi as follows.

In brief, Bi³⁺ was reduced by the photogenerated electrons to produce Bi⁰ clusters on TiO₂. Subsequently, Pd was deposited onto Bi/TiO₂ to form Pd_x/Bi/TiO₂. When the sample was exposed to the air, Bi was spontaneously oxidized into Bi₂O₃ because the Gibbs free energy of the reaction is minus. The procedure was schematically illustrated in the **Fig. 1a**.

Figure 1a. Schematic illustration of the synthetic procedures.

15) No information is provided for sample handling before XPS, XRD, or XAS measurements. This information must be provided so the reader can predict the chemical state of the catalyst and how ambient may have affected composition.

Author response: As suggested by the reviewer, we provided the information for sample handling before XPS, XRD and XAS measurements as follows.

The catalyst was activated by H₂ at 100 °C for 1 h and then cooled to room

temperature in N₂ prior to the catalytic reaction and characterizations.

16) At a minimum, the DFT model should be motivated significantly and the critical feature of surface termination be discussed directly. This is a vital feature of the model system and the electron density transfer to the metal. Maybe even add the oxygens to the bottom of the slab and show that the charge transfer still occurs.

Author response: We sincerely thank the reviewer for valuable suggestion. The reviewer is kindly referred to the reply in **comments 3 and 9**, in which we show that the charge transfer still takes place from Bi₂O₃ to Pd even if the more oxygen is added at the bottom of the slab.

Reviewer #3 (Remarks to the Author):

This paper investigates a nanometer metal/oxide interface – hybrid Pd/Bi₂O₃ – in selective hydrogenation of acetylene, with promising catalytic performances.

An original structural model for the catalytic system is proposed, which is supported by several arguments, but which can also be discussed.

1. There are several possible phases for the 1:1 BiPd IMC, for which the shortest Pd-Bi distance ranges from 2.83 Å to 2.86 Å [Canadian Mineralogist 28 (1990) 751; J. Less Common Metals 64 (1979) P17; Soviet Physics JETP 5 (1957) 1064]. However, the authors talk about the Pd_{1.0}Bi/TiO₂-IMC, suggesting that they identified a specific phase, which is unfortunately not defined with the conventional way (space group and lattice parameters, instead of PDF#29-0238). They invoked the shift in R-space (0.06 Å, Fig. S9b) to assess that the Pd-Bi distances are not the same in Pd_{1.0}/Bi₂O₃/TiO₂ and in Pd_{1.0}Bi/TiO₂-IMC. However, it does not exclude that two different IMCs phases are formed.

Author response: We agree with the reviewer that there are several possible phases for the 1:1 BiPd IMC. As noted by the reviewer, the shortest Pd-Bi distance of BiPd IMCs ranges from 2.83 Å to 2.86 Å, which is still obviously larger than that of Pd_{1.0}/Bi₂O₃/TiO₂ (2.79 Å, **Table S1**). This is confirmed by DFT calculations. As shown below in **Table S3**, we calculated four bulk PdBi IMCs from Crystallography Open

Database (<https://nanocrystallography.org/>). These IMCs all show Pd-Bi distances (2.85 – 3.03 Å) that are larger than 2.79 Å, suggesting that BiPd IMCs are not the main phase of Pd_{1.0}/Bi₂O₃/TiO₂. In this study, one of the 1:1 BiPd IMC was synthesized and took as a reference sample to illustrate the difference between BiPd IMC and Pd_{1.0}/Bi₂O₃/TiO₂. The crystal structure of PdBi/TiO₂-IMC was identified by XRD. As shown in **Fig. S9a**, PdBi/TiO₂-IMC shows characteristic diffraction peaks at 39.9 and 42.8°, corresponding to (102) and (110) of sobolevskite (hexagonal PdBi-IMC, PDF#29-0238). The space group of sobolevskite is P63/mmc(194) and the lattice parameters are a = b = 4.22 Å, c = 5.709 Å, $\alpha = \beta = 90^\circ$, and $\gamma = 120^\circ$. Pd-M (M = Pd or Bi) coordination peaks in R space of PdBi/TiO₂-IMC obviously shift (~ 0.06 Å) from that of Pd_{1.0}/Bi₂O₃/TiO₂, suggesting that PdBi IMC have longer Pd-Bi distances (**Fig. S9b**). More importantly, a massive Bi⁰ peak at 156.6/162.0 eV (**Fig. S9d**) is observed in the Bi4f of PdBi/TiO₂-IMC but is absence in that of Pd_{1.0}/Bi₂O₃/TiO₂ (**Fig. S8b**). These results clear indicate that Pd_{1.0}/Bi₂O₃/TiO₂ is composed of Pd-Bi₂O₃ hybrid clusters rather than PdBi IMC.

Table S3. DFT calculations of PdBi intermetallic from Crystallography Open Database (<https://nanocrystallography.org/>).

Composition and No. in the database	Pd-Bi CN	Pd-Bi distance /Å
PdBi No. 9004224	6	2.905
PdBi ₂ No. 9012857	8	3.029
PdBi No. 9012856	7	2.893
PdBi ₂ No. 9012855	7	2.846

As suggested by the reviewer, the following text was added into the revised manuscript.

It is frequently reported that intermetallic compounds (IMC) could be possibly formed by a so-called reactive metal-support interaction.³⁴ **However, according to the DFT calculations (Table S3), the Pd–Bi distance of PdBi IMC ranges from 2.85 – 3.03**

Å, which is obviously larger than that of Pd_{1.0}/Bi₂O₃/TiO₂ (2.79 Å, Table S2). This result excludes PdBi IMC as the main phase of Pd_{1.0}/Bi₂O₃/TiO₂. To better illustrate the difference between Pd-Bi₂O₃ nanoclusters and PdBi intermetallic compounds, we synthesized a PdBi IMC (denoted as PdBi/TiO₂) for comparison (Fig. S9). XRD patterns (Fig. S9a) of PdBi/TiO₂ exhibit characteristic peaks at 39.9 and 42.8°, corresponding to (102) and (110) of hexagonal PdBi IMC (sobolevskite, P63/mmc(194), a = b = 4.22 Å, c = 5.709 Å, α = β = 90°, and γ = 120°).

2. Information about the crystal structure of Bi₂O₃ is also lacking. There are several phases with the Bi₂O₃ stoichiometry (*Mater. Lett.* 64 (2010) 2247-2250, *Chem. Phys. Lett.* 378 (2003) 395-399, *PRB* 82 (2010) 024106, *PRB* 83 (2011) 214102, *Acta Cryst. C* 44 (1988) 587-589) but the authors do not provide any information about the phase. Experimental lattice parameters are given in the section "Computational methods", but no reference is given, suggesting that it corresponds to the measurements done by the authors. However, this is not clear for me. In addition, the choice of the (100) orientation is not really well-argued. How is it consistent with the assertion that most Pd clusters are identified to bond and hybridize with Bi₂O₃ clusters without any fixed orientation relationship or facet preference?

Author response: We thank the reviewer for raising this important issue. As noted by the reviewer and reported in the literatures (*Mater. Lett.* 64 (2010) 2247-2250; *Chem. Phys. Lett.* 378 (2003) 395-399; *Phys. Rev. B* 82 (2010) 024106; *Phys. Rev. B* 83 (2011) 214102; *Acta Cryst. C* 44 (1988) 587-589), Bi₂O₃ appears in five polymorphic modification, as α-, β-, γ-, δ-Bi₂O₃ and ε-Bi₂O₃. Among them, α-Bi₂O₃ (monoclinic, P21/c(14), a = 5.849 Å, b = 8.166 Å and c = 7.510 Å) is the room-temperature stable phase and it occurs as mineral bismite in nature. In this study, Bi₂O₃ was formed by the mild oxidation of Bi⁰ in air (2Bi(s) + 1.5O₂(g) = Bi₂O₃(s) (298–544 K), ΔG[⊖] = -574887 + 99.71T). The Gibbs free energy is minus in the temperature range of 200–544 K, which means the bismuth oxidation reaction is thermodynamically spontaneous. According to the literature (Xia et al. *Transactions of Nonferrous Metals Society of China*, 2012, 22, 2289-2294), the mild oxidation of Bi⁰ always forms α-Bi₂O₃

(monoclinic, P21/c(14)). In this study, the crystal structure of α -Bi₂O₃ was identified by ADF-STEM, as shown in **Figs. 1b**. Specifically, the structural projection and Z²-map of ordered Bi₂O₃ feature a wave-shaped arrangement of Bi atomic columns from the [100] direction, which match well with the atomic structure observed by ADF-STEM (**Fig. 1b**).

We have tried to identify the orientation of Pd bonding with Bi₂O₃ clusters by ADF-STEM (**Fig. 1d and S1c**). However, due to the ultra-small size and large strain of the clusters, most Pd clusters are identified to bond and hybridize with Bi₂O₃ clusters without any fixed orientation relationship or facet preference. We chose the (100) surface in the calculations because of several reasons. Firstly, (100) surface is a low-index surface that normally has large proportions among the overall crystal surfaces, which makes this surface representative. Secondly, the Bi₂O₃(100) surface has a layered structure, and the Bi-terminated surface can be exposed under reducing conditions. We have also demonstrated the stability of Bi-terminated Bi₂O₃(100) surface with thermodynamic analyses regarding surface reduction. *See the reply to comment 9) of referee 2*. Thirdly, we found that only such Bi-terminated Bi₂O₃(100) surface can give rise to consistent Pd-Bi distances with those determined experimentally. In addition, we believe that the charge transfer from Bi₂O₃ to Pd is the key of the high reactivity of Pd/Bi₂O₃ catalysts, and the negatively charged Pd atoms on Bi₂O₃ behaves differently with respect to Pd substance. We have shown in computations that the acetylene hydrogenation reaction takes place on Pd sites rather than at Bi₂O₃ site, and hence the reactivity behavior of Pd-cluster during acetylene hydrogenations is possibly independent with surface orientations. We have added discussion in the manuscript to clarify this point.

As suggested by the reviewer, we have added more discussion into the revised manuscript and supporting information to illustrate the structure of Bi₂O₃ and the choice of (100) orientation.

Interestingly, close inspection of individual Bi cluster on the TiO₂ support by using aberration-corrected annular dark field scanning transmission electron microscopy (ADF-STEM) indicates that the Bi cluster has an ordered α -Bi₂O₃ structure with a

highly distorted lattice, which exhibits relatively weak diffuse peaks in the FFT (**Fig. 1b**). This is consistent with the literature result, which indicates that the mild oxidation of Bi⁰ in the air is thermodynamically spontaneous and usually forms monoclinic α -Bi₂O₃ (P21/c(14), a= 5.849 Å, b= 8.166 Å and c= 7.510 Å).

3. According to the experimental measurements, the Bi-Pd distance is larger in Pd_{1.0}Bi/TiO₂-IMC than in Pd_{1.0}/Bi₂O₃/TiO₂. If the assumptions made about the structures of Pd_{1.0}/Bi₂O₃/TiO₂ and Pd_{1.0}Bi/TiO₂-IMC are correct, different Bi-Pd distances calculated by DFT, using the model already built for Pd_{1.0}/Bi₂O₃/TiO₂ and a model that the authors can build for Pd_{1.0}Bi/TiO₂-IMC, based on the crystal structure determined by XRD, could support this assumption. Such a calculation would also provide information about the charge transfer in the Pd_{1.0}/Bi₂O₃/TiO₂ system and in the model built for Pd_{1.0}Bi/TiO₂-IMC, which can be useful to strengthen the analysis of the spectroscopic data.

Author response: As suggested by the reviewer, we built the model of Pd_{1.0}Bi/TiO₂-IMC based on the crystal structure determined by XRD (i.e., Sobolevskite, PDF#29-0238). The space group of the hexagonal PdBi-IMC is P63/mmc(194) and the lattice parameters are a = b = 4.22 Å, c = 5.709 Å, $\alpha = \beta = 90^\circ$, and $\gamma = 120^\circ$. Upon DFT optimization, the Pd-Bi distance is calculated to be 2.91 Å (**Table S3**), which is obviously larger than the distance of 2.75 Å in the Pd₈-cluster model. These results are consistent with the EXAFS results. As shown in Fig. **S9b**, Pd-M (M = Pd or Bi) coordination peaks in R space of PdBi/TiO₂-IMC obviously shift (~ 0.06 Å) from that of Pd_{1.0}/Bi₂O₃/TiO₂, suggesting that PdBi IMC have longer Pd-Bi distances. In addition, we also calculated the Bader charge of Pd atoms on the PdBi-IMC model. Interestingly, Pd atoms in PdBi IMC model are more electron-rich (average charge of -0.36 e) than those in Pd-Bi₂O₃ hybrid clusters model (-0.21 e), which is well consistent with downshift of Pd3d XPS peaks shown in **Fig. S9c** and the weaker Pd L3 edge XANES peak in **Fig. S10**. The above results suggests that the assumptions made about the structures of Pd_{1.0}/Bi₂O₃/TiO₂ and PdBi/TiO₂-IMC are correct.

As suggested by the reviewer, we added the following discussion into the revised

manuscript.

The Pd-Bi pair distribution function of the Pd₈ cluster structure is shown in **Fig. S12**. In this figure the dominant peak appears at ~2.75 Å, which is smaller than that in the PdBi intermetallic model (~ 2.91 Å, **Table S3**). In addition, Bader charge analysis suggests that Pd atoms in PdBi IMC model are more electron-rich (average charge of -0.36 e) than those in Pd-Bi₂O₃ hybrid clusters model (-0.21 e). These results are consistent with Pd K-edge EXAFS and Pd3d XPS shown in **Fig. S9**, which therefore validate the reliability of the Pd₈-cluster model.

4. *Fig. 4b provides a reaction path for the hydrogenation reaction, (i) on the surface the Pd_{1,0}/Bi₂O₃/TiO₂ model provided by the authors and (ii) on Pd(111). The values of the barriers are given for Pd_{1,0}/Bi₂O₃/TiO₂ in the supplemental materials (Tab. S3), but not for Pd(111). The latter could have been compared with the literature, to assess the reliability of the computational method (constrained minimization method to locate the transition state structures).*

Author response: As suggested by the reviewer, we compared the barriers of acetylene hydrogenation to ethane on Pd(111) that were reported by different groups and obtained with different transition state searching methods. As shown in **Table S5**, the calculated barriers in the current work are generally consistent with reported values, demonstrating that our calculated results are reliable. In addition, we further calculated the vibrational frequency based on the transition state structures on Pd(111). All the transition states were characterized to possess only one imaginary frequency along the bond formation of hydrogenation reactions. These results further demonstrate the reliability of the constrained minimization method that we used in this work.

In addition to revised **Table S5**, we also added the following discussion into the manuscript.

The calculated barriers are generally consistent with reported values (**Table S5**), demonstrating that our calculated results are reliable. In addition, we further calculated the vibrational frequency based on the transition state structures on Pd(111). All the transition states were characterized to possess only one imaginary frequency along the bond formation of hydrogenation reactions. These results further demonstrate the

reliability of the constrained minimization method that we used in this work.

Table S5. Energy barriers of acetylene hydrogenation to ethane over Pd(111). The unit of energy is in eV.

Reaction	Barrier in this work /eV	Imaginary frequencies /cm ⁻¹	Barrier in literatures /eV
CHCH→CHCH₂	0.88	74.48	0.96 ^{a,c}
			0.55 ^b
			1.04 ^d
CHCH₂→CH₂CH₂	0.85	97.10	0.89 ^a
			0.77 ^b
			0.97 ^c
			0.93 ^d
CH₂CH₂→CH₂CH₃	0.88	79.88	0.91 ^a
			0.88 ^b
CH₂CH₂→CH₃CH₃	0.65	119.00	0.57 ^b

a: Journal of Catalysis 305, 2013, 264–276

b: Science, 320, 5881, 2008, 1320-1322

c: J. Phys. Chem. C 2010, 114, 17683-17692

d: J. Mol. Catal. A Chem., 2011, 344, 37-46

REVIEWERS' COMMENTS

Reviewer #1 (Remarks to the Author):

The authors have addressed my concerns. I suggest publication.

Reviewer #2 (Remarks to the Author):

Review Summary:

I believe that the authors have sufficiently addressed all of my original comments. My initial reservations have been laid to rest by their responses and additional focused studies.

Reviewer #3 (Remarks to the Author):

The modifications made by the authors are adequate.

A few comments/typos

The label for space groups are $P6_3/mmc$ (3 = subscript).

It is the same for $P2_1/c$. The P in space group symbol should be placed in italic font.

Line 254 (clear)

Line 431 (absence)

Fig 4b Pd8 cluster -> Pd8cluster supported on Bi2O3(100)

Fig 4c: it is not clear for me what is the physical model that supports the fit by the blue line. Suggestion: indicate that it is a guide for the eyes.

General response: We sincerely thank the editor and all reviewers for their valuable feedback that we have based on to improve the quality of our manuscript. The reviewer comments are listed below in *italicized font* and specific concerns have been numbered. Our response is given point-by-point in normal font and changes/additions to the manuscript are given in **red text**.

REVIEWER COMMENTS

Reviewer #1 (Remarks to the Author):

The authors have addressed my concerns. I suggest publication.

Author Reply: Thank you so much for your kind review and favorable comments.

Reviewer #2 (Remarks to the Author):

I believe that the authors have sufficiently addressed all of my original comments. My initial reservations have been laid to rest by their responses and additional focused studies.

Author Reply: Thank you so much for your kind review and favorable comments.

Reviewer #3 (Remarks to the Author):

The modifications made by the authors are adequate.

Author Reply: Thank you so much for your kind review and favorable comments.

A few comments/typos

1. *The label for space groups are $P6_3/mmc$ ($3 = \text{subscript}$). It is the same for $P2_1/c$.*

The P in space group symbol should be placed in italic font.

Author Reply: In the revised manuscript, $P6_3/mmc$ and $P2_1/c$ are changed to **$P6_3/mmc$** and **$P2_1/c$** , respectively.

2. *Line 254 (clear)*

Author Reply: We sincerely thank the reviewer for careful reading and valuable comments. We have carefully read the manuscript and tried our best to polish the

language. In the revised manuscript, “clear” is changed to “clearly”.

3. *Line 431 (absence)*

Author Reply: As suggested by the reviewer, we have changed “absence” to “absent”.

4. *Fig 4b Pd8 cluster -> Pd8cluster supported on Bi2O3(100)*

Author Reply: We have revised Fig. 4b according to the reviewer’s suggestion.

5. *Fig 4c: it is not clear for me what is the physical model that supports the fit by the blue line. Suggestion: indicate that it is a guide for the eyes.*

Author Reply: As suggested by the reviewer, we add the following text into the manuscript to indicate that the blue line is a guide for the eyes.

The blue fitted line is a guide for the eyes. It shows that a more negative ϵ_d corresponds to a more positive E_{ads} of C_2H_4 .